# Interfacial water confers transcription factors with dinucleotide specificity

Ekaterina Morgunova [1], Gabor Nagy [2], Yimeng Yin [3,4], Fangjie Zhu [5], Sonali Priyadarshini Nayak[2,6], Tianyi Xiao[3], Ilya Sokolov[3], Alexander Popov [7], Charles Laughton [8], Helmut Grubmuller [2] & Jussi Taipale [1,3,9,10] ✉

Transcription factors (TFs) recognize specific bases within their DNA-binding motifs, with each base contributing nearly independently to total binding energy. However, the energetic contributions of particular dinucleotides can deviate strongly from the additive approximation, indicating that some TFs can specifically recognize DNA dinucleotides. Here we solved high-resolution (<1 Å) structures of MYF5 and BARHL2 bound to DNAs containing sets of dinucleotides that have different affinities to the proteins. The dinucleotides were recognized either enthalpically, by an extensive water network that connects the adjacent bases to the TF, or entropically, by a hydrophobic patch that maintained interfacial water mobility. This mechanism confers differential temperature sensitivity to the optimal sites, with implications for thermal regulation of gene expression. Our results uncover the enigma of how TFs can recognize more complex local features than mononucleotides and demonstrate that water-mediated recognition is important for predicting affinities of macromolecules from their sequence.

Proteins can bind to DNA in a non-sequence-specific manner by binding to its backbone or recognize specific sequences by interactions with DNA bases. The DNA sequence can be directly read out by a TF through hydrogen bonding and/or hydrophobic interactions between protein amino acids and DNA bases[1–4]. Sequences can also be read indirectly through effects on DNA structure and shape[4–8] and by water-mediated contacts between TFs and DNA bases[7,9–11].

Direct interactions are relatively insensitive to the local environment and result in the recognition of individual mononucleotides, with total binding energy being approximately additive[6,10,12–17]. This additive model holds very well for the majority of base positions across a broad range of structural TF families but often performs less well in predicting energy contributions of adjacent DNA bases and bases that

are recognized indirectly[18]. Owing to the influence of adjacent bases on each other, TF binding to DNA can be better approximated by models that use dinucleotide instead of mononucleotide features[1,6,8,18–20]. This improved performance is explained by the fact that dinucleotide content affects the local dynamics and structure of the DNA backbone and the relative orientation of the bases[21]. For example, nucleosomes bend DNA and preferentially bind to DNA sequences where particular dinucleotides are located at specific positions relative to the direction of the DNA bending[22–24]. Some TFs can also affect the conformation of the DNA backbone[25–32], and their specificity towards dinucleotides could similarly be explained by the contribution of the dinucleotides to the structure and flexibility of the DNA backbone. However, given that the bending energy of nucleosomal DNA is only 0.14 kcal mol⁻¹

[1]Department of Medical Biochemistry and Biophysics, Karolinska Institutet, Stockholm, Sweden. [2]Theoretical and Computational Biophysics Department, Max Planck Institute for Multidisciplinary Sciences, Göttingen, Germany. [3]Department of Biochemistry, University of Cambridge, Cambridge, UK. [4]State Key Laboratory of Cardiovascular Diseases and Medical Innovation Center, Shanghai East Hospital, School of Medicine, Tongji University, Shanghai, China. [5]Haixia Institute of Science and Technology, Fujian Agriculture and Forestry University, Fuzhou, Fujian, China. [6]Max Planck School Matter to Life, Heidelberg, Germany. [7]ESRF Grenoble, Grenoble, France. [8]School of Pharmacy and Biodiscovery Institute, University of Nottingham, Nottingham, UK. [9]Applied Tumor Genomics Research Program, Faculty of Medicine, University of Helsinki, Helsinki, Finland. [10]Generative and Synthetic Genomics Programme, Wellcome Sanger Institute, Hinxton, UK. ✉e-mail: jussi.taipale@ki.se

per base pair[33,34], the structural distortion caused by histone octamer or DNA-bending TFs can only result in relatively weak (less than two-fold) local dinucleotide preferences[35–37] and thus can not account for the highly specific recognition of dinucleotides observed for many TFs. Furthermore, many TFs with preferences for particular dinucleotides can bind to DNA without inducing major changes to its canonical B-form, suggesting the existence of additional mechanisms for the recognition of dinucleotides.

Previously, we have shown that binding of TFs to two distinct sequence optima can be caused by partial independence of the two contributions, ΔH and TΔS, to binding free energy, with different DNA sequences representing enthalpic and entropic optima for binding[38]. We also proposed that such a thermodynamic mechanism, resembling entropy–enthalpy compensation[39–41], affects many other macromolecular interactions. To investigate the molecular interactions that contribute to the entropic and enthalpic optima and enable TFs to specifically recognize dinucleotides, in this work we have characterized the structures of MYF5 and BARHL2 bound to multiple different optimal and sub-optimal DNA sequences.

## Results

### Mechanism of MYF5 binding to dinucleotides

To investigate how basic helix–loop–helix (bHLH) proteins can recognize dinucleotides flanking the E-box core sequences and bind to two distinct DNA sequences with similar affinity[18,38,42–45], we solved the structure of the bHLH domain of the MYF5 homodimer bound to two different DNA fragments representing the entropic and enthalpic[38] optima (Fig. 1a,b and Table 1). One sequence is palindromic and contains an E-box-like core, CAGCTG[46], flanked by the entropically optimal GT (**GT**CAGCTG**AC**, with divergent dinucleotides shown in bold; hereafter MYF5–DNA$^{GT}$), whereas the other sequence, **AA**CAGCTG**AC** (hereafter MYF5–DNA$^{AA}$) is non-palindromic, containing one enthalpic half-site (**AA**CAG) and one entropic half-site (CTG**AC**, the reverse complement of **GT**CAG).

The structures were solved at 3.1 Å and 2.3 Å resolution using X-ray crystallography (see Methods). The overall fold of the bHLH domain is strikingly similar to that observed in other members of this family, including MyoD, MAX and c-Myc (Extended Data Fig. 1). Each MYF5 monomer forms two long helices connected by a nine-residue loop. The first helix (H1) contains the basic region that binds to DNA, whereas the second helix (H2) is responsible for dimerization[47] (Fig. 1a). As observed in all known bHLH-DNA complexes, the basic region fits into the major groove without significantly bending the DNA (Fig. 1a–e and Extended Data Fig. 1).

To determine whether strong binding to the two optimal sequences can be explained by their similar DNA backbone shape and the positions of the charged atoms, we performed a comparative analysis of the DNA shape with w3DNA[48] and DNAphi[49] (Supplementary Table 1 and Extended Data Fig. 2). This analysis revealed that although the DNA backbone shapes of both the MYF5–DNA$^{GT}$ and MYF5–DNA$^{AA}$ are similar to canonical B-DNA (Fig. 1d), the minor groove widths and calculated electrostatic potentials are markedly different between the GT and AA DNAs (Extended Data Fig. 2). Furthermore, the positioning of charged atoms in the major grooves of MYF5–DNA$^{GT}$ and MYF5–DNA$^{AA}$ are clearly different (Fig. 1e), suggesting that similarity of DNA backbone shape and charge distribution cannot explain the preference of MYF5 to flanking AA and GT dinucleotides.

In both MYF5 complexes, only two direct contacts are formed between the core E-box (CAGCTG) sequence and the MYF5 protein: the side-chain oxygen of Glu92 of one MYF5 monomer hydrogen-bonds to nitrogen 4 (N4) of the first cytosine (C$_7$) and a symmetric contact is made by the other MYF5 monomer (Fig. 1h). In addition to the specific base contact, DNA affinity is increased by non-sequence-specific hydrogen bonds formed by Arg85, Thr89, Arg93 and Arg95, Asn100 and Lys120 with the ribose oxygens and phosphate moieties. The AA

flank in DNA$^{AA}$ is recognized by Arg91, which contacts the first A (A$_5$) through a water molecule (Fig. 1g and Extended Data Fig. 1c–e). However, in DNA$^{GT}$, Arg91 does not interact with water but instead makes direct hydrogen bond contact with the G$_5$ (Fig. 1h).

These results suggest that the dual specificity of MYF5 is not a result of similarities in DNA shape or charge distribution between the recognized dinucleotides. Instead, the two high-affinity sites appear to result from the entropic and enthalpic optima[38]; in the entropic optimum, the G in the GT dinucleotide is bound directly, freeing a water molecule, whereas in the enthalpic optimum, the first A of the AA dinucleotide is bound by a fixed water molecule. However, because of the relatively low resolution of the MYF5 structures, we were unable to determine why the G-bound state prefers an adjacent T while the A-bound state prefers an adjacent A.

### Mechanism of BARHL2 binding to dinucleotides

To study TF binding to dinucleotides in more molecular detail, we performed an extensive structural analysis of DNA binding by the homeodomain TF BARHL2. BARHL2 can recognize a canonical homeobox-like sequence (TAA**TT**G) but, unlike most other homeodomain proteins, can also bind with even higher affinity to TAA**AC**G, which contains a different dinucleotide, AC. To investigate BARHL2 specificity at the region containing the dinucleotides, we solved BARHL2 structures bound to eight different DNAs, including the enthalpic (TAA**AC**G) and entropic (TAA**TT**G) optima, two more weakly bound sequences that are within one substitution from both optima (TAA**AT**G and TAA**TC**G) and four sub-optimal sequences (TAA**TG**G, TAA**GT**G, TAA**GC**G, TAA**CC**G). The structures were of high to extremely high resolution (from 2.6 Å to 0.95 Å, Table 1), enabling detailed analysis of the interactions contributing to the binding affinity of each sequence. For clarity, we refer to the complexes using the variable dinucleotides hereafter (for example TAA**AC**G is DNA$^{AC}$). One BARHL2–DNA$^{AC}$ crystal diffracted to 0.95 Å, which is thus far the highest resolution for any TF–DNA complex. The high resolution revealed alternative conformations for most of the phosphates, sugars and bases of the DNA backbone (Fig. 2a and Extended Data Fig. 3), hitherto only observed in two previous TF–DNA complexes[50,51].

Analysis of all the complexes confirmed that BARHL2 exhibits common homeobox features and encompasses a comparatively long, positively charged amino-terminal peptide and three α-helices connected by short loops (Fig. 2b–e and Extended Data Fig. 4). In all solved homeodomain–DNA complexes, the last A of the TA**A** sequence is recognized by an asparagine (Asn282 in BARHL2), which forms a hydrogen bond with A$_6$ (T$_4$A$_5$A$_6$). In BARHL2, Thr278 also forms a hydrophobic contact with carbon atom C8 of the aromatic ring of the same adenine. The TAA sequence is also recognized by Arg233 and Arg236 via the minor groove, forming direct and water-mediated hydrogen bonds with all three base pairs (Fig. 2e). These results indicate that within the conserved TAA sequence, the interactions between BARHL2 and DNA are typical of those seen for the characterized anterior-type homeobox TFs.

We next analyzed the molecular basis of the sequence specificity of BARHL2 towards the dinucleotides following the TAA motif. First, we checked whether the high affinity of the two distinct optimal sequences DNA$^{AC}$ and DNA$^{TT}$ can be explained by a similarity in DNA backbone shape or electrostatic potential. However, DNAs in all structures had almost canonical B-shapes (for example, DNA$^{TT}$; Fig. 2f). Comparison of the DNAs between the DNA$^{AC}$, DNA$^{TT}$, DNA$^{AT}$ and DNA$^{TC}$ revealed that the largest shift from B-shape was observed at the position of the variable dinucleotide between DNA$^{AC}$ and DNA$^{TT}$ (2.9 Å; Fig. 2g,h). Thus, the two high-affinity sequences were farther apart from each other than from the two lower-affinity sequences. Furthermore, calculated minor groove widths and electrostatic potentials of DNA$^{AC}$ and DNA$^{TT}$ were also markedly different (Extended Data Fig. 2), suggesting that the similarity of their affinity to BARHL2 is not caused by similarities in DNA backbone shape, minor groove width or electrostatic potential.

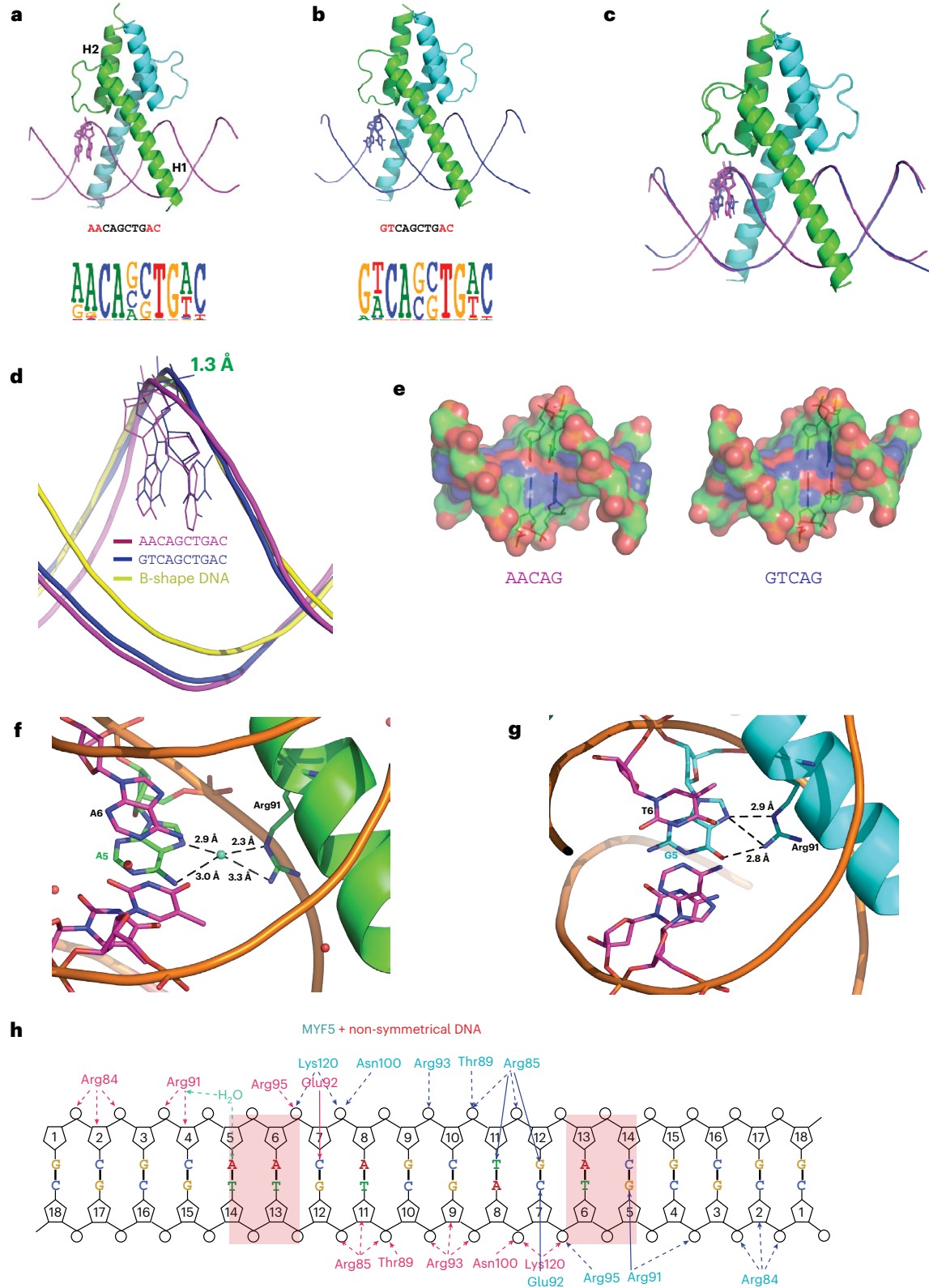

**Fig. 1 | Structures of MYF5–DNA complexes. a,b**, Overview of two structures of MYF5 DNA-binding domains bound to non-symmetrical DNA **AA**CAGCTG**AC** (**a**) and symmetrical DNA **GT**CAGCTG**AC** (**b**). Logos of the motifs of the homologous protein MYF6 are shown below the structures. Note that MYF6 prefers to bind to either the AA or GT flank before the E-box. **c**, Structural alignment of two MYF5 structures containing different DNA sequences. The symmetrical DNA **GT**CAGCTG**AC** is in magenta; the non-symmetrical DNA **GT**CAGCTG**TT** is in blue. **d**, Enlarged view of parts of DNA, showing the largest deviation of experimental DNA from B-shape DNA (shown in yellow). The measured distance between the corresponding phosphorus atoms is shown as green dashed lines. **e**, Charge distribution on the surface of the region containing the divergent dinucleotide (blue is positive, red is negative). Note that charge distribution is very different at the region of the GT and AA dinucleotides. **f**, A water molecule mediates the contact between Arg91 and $A_5$ of the AA dinucleotide. **g**, Direct contact is formed between Arg91 and $G_5$ of the GT dinucleotide. **h**, Schematic representation of the contacts formed by MYF5 with non-symmetrical DNA. Left and right sides show contacts to AA and GT, respectively.

**Table 1 | X-ray data collection and refinement statistics**

| Protein/core | MYF5/CAGCTG | | BARHL2/TAA | | | | | | | | |
|---|---|---|---|---|---|---|---|---|---|---|---|
| Flanking DNA | AC/GT | AA/GT | AC | AC | TT | AT | TC | GC | CC | TG | GT |
| **Data collection** | | | | | | | | | | | |
| Space group | C 2 | C 2 | $P2_12_12_1$ | $P2_12_12_1$ | $P 2_1$ | $P 2_1$ | $P 2_1$ | $P 2_1$ | $P 2_1$ | $P 2_12_12_1$ | $P 2_1$ |
| Cell dimensions | | | | | | | | | | | |
| a, b, c (Å) | 166.7, 33.9, 53.7 | 170.4, 33.8, 53.6 | 38.8, 47.3, 72.5 | 38.6, 47.1, 72.3 | 39.6, 123.1, 62.2 | 39.0, 134.3, 65.6 | 39.6, 123.8, 62.7 | 39.1, 133.7, 64.9 | 39.4, 134.5, 64.5 | 38.8, 46.8, 71.8 | 39.6, 123.7, 62.4 |
| α, β, γ (°) | 90, 91.04, 90 | 90, 91.9, 90 | 90, 90, 90 | 90, 90, 90 | 90, 94.2, 90 | 90, 92.6, 90 | 90, 93.5, 90 | 90, 92.7, 90 | 90, 93.2, 90 | 90, 90, 90 | 90, 93.9, 90 |
| Resolution (Å)[a] | 41.67,3.0 (3.10–3.0) | 46.00,2.28 (2.36–2.28) | 28.78,0.95 (0.98–0.95) | 29.89,1.30 (1.35–1.30) | 39.49,1.85 (1.91–1.85) | 46.91,2.4 (2.48–2.4) | 39.55,1.7 (1.76–1.7) | 39.04,2.1 (2.17–2.1) | 39.37,2.6 (2.69–2.6) | 35.88,1.45 (1.50–1.45) | 39.43,2.05 (2.12–2.05) |
| R-merge | 0.063 (1.062) | 0.024 (0.72) | 0.045 (2.89) | 0.073 (1.96) | 0.071 (2.70) | 0.145 (2.71) | 0.050 (1.77) | 0.11 (4.61) | 0.19 (1.54) | 0.077 (1.41) | 0.153 (2.00) |
| I/σI | 8.46 (1.08) | 17.5 (1.8) | 15.3 (0.5) | 10.2 (0.7) | 8.5 (0.4) | 8.8 (0.7) | 9.5 (0.5) | 7.0 (0.3) | 5.9 (1.0) | 13.6 (0.6) | 7.7 (0.8) |
| Completeness (%) | 90.1 (93.2) | 94.1 (97.2) | 99.5 (97.9) | 99.5 (97.2) | 99.3 (97.7) | 99.03 (100) | 99.3 (97.2) | 99.6 (99.8) | 99.3 (96.9) | 99.2 (94.4) | 99.6 (98.7) |
| Redundancy | 2.8 (2.7) | 5.0 (5.2) | 6.5 (6.0) | 7.1 (6.6) | 3.8 (3.7) | 7.2 (6.6) | 3.8 (2.9) | 5.7 (5.7) | 3.8 (4.0) | 7.9 (3.3) | 5.2 (4.8) |
| **Refinement** | | | | | | | | | | | |
| Resolution (Å) | 3.0 | 2.28 | 0.95 | 1.3 | 1.85 | 2.4 | 1.7 | 2.1 | 2.6 | 1.45 | 2.05 |
| No. of reflections | 5,673 | 14,060 | 83,944 | 32,821 | 50,206 | 26,124 | 65,719 | 38,705 | 20,488 | 23,721 | 37,295 |
| $R_{work}$ / $R_{free}$ | 0.23/0.28 | 0.24 /0.22 | 0.14 /0.17 | 0.15/0.20 | 0.23/0.26 | 0.23 /0.28 | 0.22/0.25 | 0.24 /0.27 | 0.24 /0.27 | 0.18/0.22 | 0.23/0.26 |
| No. of atoms | 1,683 | 1,779 | 1,937 | 1,307 | 4,384 | 4,884 | 4,472 | 4,148 | 4,572 | 1,362 | 4,427 |
| Protein and DNA | 1,680 | 1,701 | 1,507 | 1,041 | 4,053 | 4,557 | 4,035 | 4,027 | 4,398 | 1,128 | 4,076 |
| Water | 3 | 78 | 430 | 266 | 331 | 327 | 437 | 121 | 174 | 234 | 351 |
| B factors | 106.02 | 84.13 | 17.59 | 24.03 | 45.18 | 48.82 | 42.33 | 65.87 | 56.94 | 23.28 | 43.38 |
| R.M.S.D. | | | | | | | | | | | |
| Bond lengths (Å) | 0.016 | 0.005 | 0.018 | 0.015 | 0.013 | 0.014 | 0.012 | 0.015 | 0.015 | 0.016 | 0.016 |
| Bond angles (°) | 1.66 | 1.11 | 2.43 | 1.92 | 1.69 | 1.78 | 1.71 | 2.28 | 2.17 | 2.20 | 2.35 |

[a]Statistics for the highest-resolution shell are shown in parentheses.

To investigate whether the dinucleotides would instead be recognized by networks of water molecules, we studied the arrangement of water molecules in the protein–DNA interfaces. We first assessed whether differences in resolution would affect the detection of the water molecules using the two BARHL2–DNA$^{AC}$ structures solved at 0.95 Å and 1.3 Å. Although overall more water molecules are seen in the protein–DNA interface in the 0.95 Å structure, inspection of the water content at the site of the variant dinucleotide at both resolutions showed conserved positions of all of the ten water molecules, indicating that at this range, resolution does not substantially affect detection of water molecules contributing to recognition of the dinucleotide (Extended Data Fig. 3g).

We then compared the water arrangements and molecular interactions between BARHL2 and the eight different DNA sequences (Fig. 3). Analysis of the highest-affinity complex, BARHL2–DNA$^{AC}$, revealed that the high-affinity binding of BARHL to DNA$^{AC}$ is caused by a combination of indirect and direct recognition of DNA. Key amino acids involved in the recognition of the AC dinucleotide and its complementary bases are Asn282 and two threonines: Thr278 and Thr285. Thr278 is connected to DNA by two water chains (Fig. 3a). The first water chain contains three water molecules that together connect the side-chain oxygen of Thr278 with the side-chain oxygen of Asn282 and N7 of the A of the AC dinucleotide (A$_7$). On the other side, Thr278 is connected to N4 of the C of AC (C$_8$) by a second chain of four water molecules. Thus, Thr278 orders water molecules in such a way that they contact both bases of the AC dinucleotide. The two water chains are also connected by one water molecule, which is within a hydrogen-bond distance from N6 of A$_7$ and O6 of G$_{17}$ opposite to C$_8$ of AC. The T$_{18}$ opposite to the A$_7$ of the AC dinucleotide is also recognized directly by a hydrophobic contact to the methyl group (CG2) of Thr285 (Fig. 3a). In addition, the side-chain oxygen of Thr285 contacts N7 of the G$_{17}$ opposite to the AC dinucleotide through a water molecule that is a part of the second water chain (Fig. 3a). Furthermore, the T$_{18}$ opposite to the A$_7$ of AC is connected through a water molecule to the side-chain oxygens of Asn282 and Thr285. This water molecule is in a network that includes another water that in turn is connected to N7 of G$_{17}$ opposite to C$_8$ of AC. Taken together, these results show that DNA$^{AC}$ is bound by a combination of a direct hydrophobic interaction and an extensive water network between the protein and DNA (Fig. 3a).

The molecular interactions observed in the structure of BARHL2 bound to DNA$^{TT}$, which showed the next highest affinity in the SELEX experiments, were strikingly different from those of BARHL2–DNA$^{AC}$. The extensive water network is lost because of a hydrophobic patch in the major groove, formed by the methyl groups of the two Ts (Fig. 3b). The hydrophobic contact with Thr285 is replaced by hydrophobic contact between the methyl group (CG2) of Thr278 and C$_7$ of the first T (T$_7$) of TT. Only one water connecting the side-chain oxygen of Thr285 with the side-chain oxygen of Asn282 was observed in the region around Thr285. On the other side, Asn282 is connected to O4 of T$_7$ through one water and to O4 of the second T (T$_8$) of TT through a water chain containing three molecules. Thus, most of the contacts in the complex BARHL2–DNA$^{TT}$ are shifted from the Thr285 region to the Thr278. An additional observation is that the side chain of Arg289 forms a

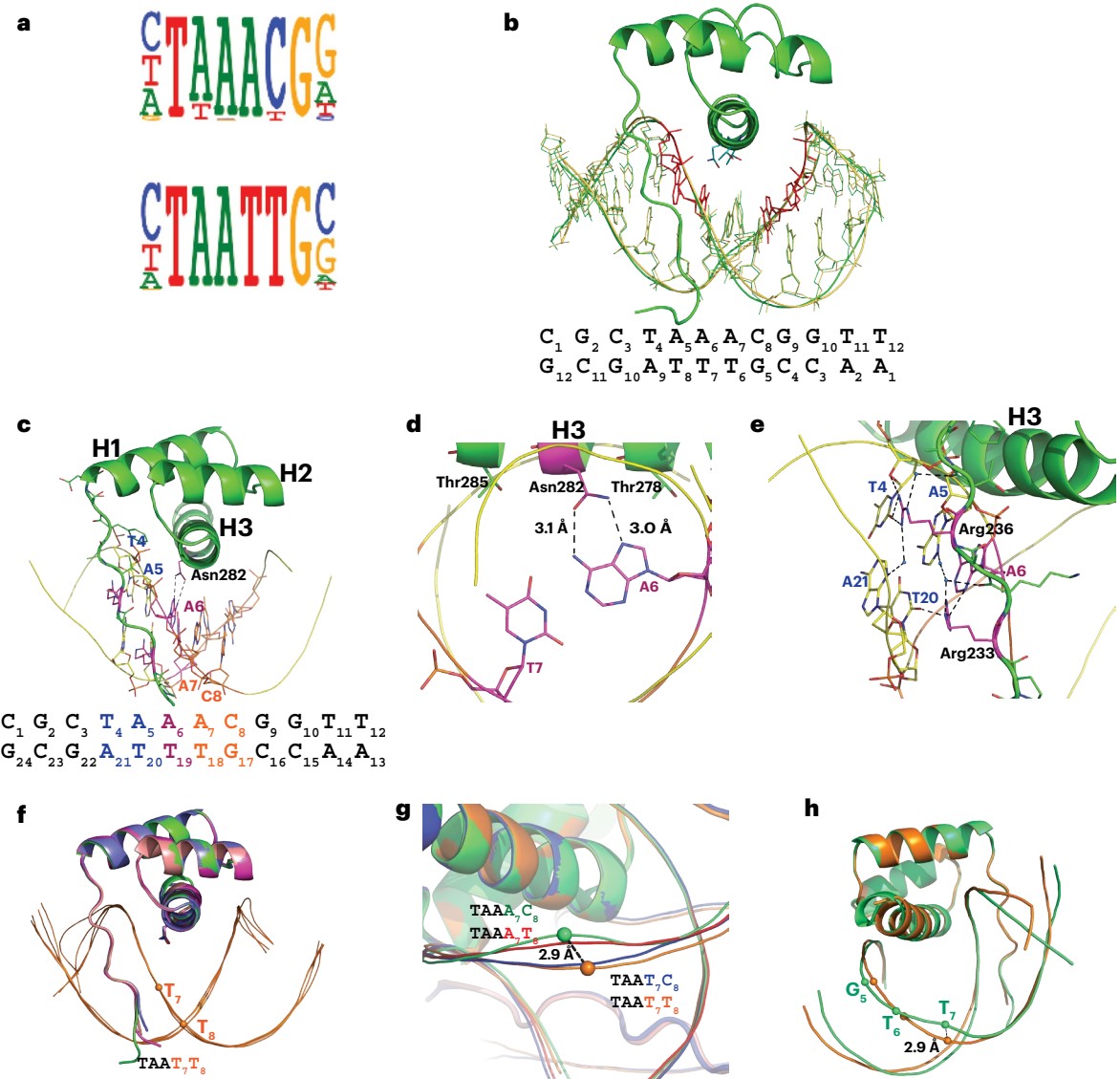

**Fig. 2 | Structures of BARHL2–DNA complexes. a**, Sequence logos for the two distinct DNA-binding specificities of the BARHL2 protein. Note that BARHL2 binds to the dinucleotide after the TAA sequence, preferring either AC or TT. **b**, Overall structure of BARHL2 bound to the DNA $C_1G_2C_3T_4A_5A_6A_7C_8G_9G_{10}T_{11}T_{12}$ (BARHL2–DNA$^{AC}$). The two conformations of the DNA backbone phosphates, sugars and bases observed in the ultra-high-resolution structure (0.95 Å) are indicated in green and yellow. The bases that only show a single conformation, $A_5A_6$ of the forward strand and $C_4G_5$ of the complementary strand, are presented as ball-and-stick models in red. The sequence of the double-stranded DNA is shown below. **c,d**, BARHL2 interacts with the TAA part of its recognition sequence using a similar mechanism as other homeodomains (**c**); close-up view of the interaction between $A_6$ (magenta) and the canonical Asn282 (**d**). The divergent dinucleotides are in orange. **e**, Close-up view of the interactions via the minor groove. The $T_4A_5$ sequence (blue) is bound by minor groove contacts formed between the amino-terminal tail residues Arg233 and Arg236. **f**, Structural alignment of four subunits of the complex BARHL2–DNA$^{TT}$. The largest shift for DNA strands between four subunits is <1 Å. **g**, Alignment of four complexes with different DNAs. Color codes are red, DNA$^{AT}$; blue, DNA$^{TC}$; orange, DNA$^{TT}$; green, DNA$^{AC}$. The corresponding DNA sequences are written using the color code of the figure. **h**, Distance between the DNA backbones in the part of divergent dinucleotides of DNA$^{TT}$ and DNA$^{AC}$ is 2.9 Å.

hydrophobic contact with the methyl group of Thr285, in contrast to BARHL2–DNA$^{AC}$, in which Arg289 shows two conformations, both of which are turned away from Thr285 (Fig. 3b).

To further examine the role of water molecules in the recognition of DNA$^{AC}$ and DNA$^{TT}$ by BARHL2, we performed crystal lattice molecular dynamics simulations of BARHL2–DNA$^{AC}$ and BARHL2–DNA$^{TT}$. The number of low-mobility water molecules observed at the protein–DNA interface was significantly greater for BARHL2–DNA$^{AC}$ than BARHL2–DNA$^{TT}$. To check the impact, if any, of the different packing arrangements of the complexes in the two different crystal lattices (P $2_12_12_1$ for BARHL2–DNA$^{AC}$, P $2_1$ for BARHL2–DNA$^{TT}$), simulations were additionally performed on BARHL2–DNA$^{AC}$ after in silico mutation of the sequence to DNA$^{TT}$. A significant reduction in the number of

low-mobility waters was observed. However, simulations on BARHL2–DNA$^{TT}$ after in silico mutation to DNA$^{AC}$ did not show a significant change in the reverse direction (Fig. 3j; see Methods for details). These results suggest that the creation of high-affinity water binding sites requires both the 'right' DNA sequence and an ability to adopt the 'right' conformation, which (artificially) embedding in a sub-optimal crystal lattice can impede.

The ability of BARHL2 to specifically recognize dinucleotides is shown by the fact that it has the highest affinity to DNA$^{AC}$ and DNA$^{TT}$, sequences that are separated by two base substitutions (Hamming distance of two). To understand why both possible single base substitutions of AC that make it more similar to TT (AT and TC) lead to a decrease in affinity, we first investigated why DNA$^{AT}$ has a lower affinity

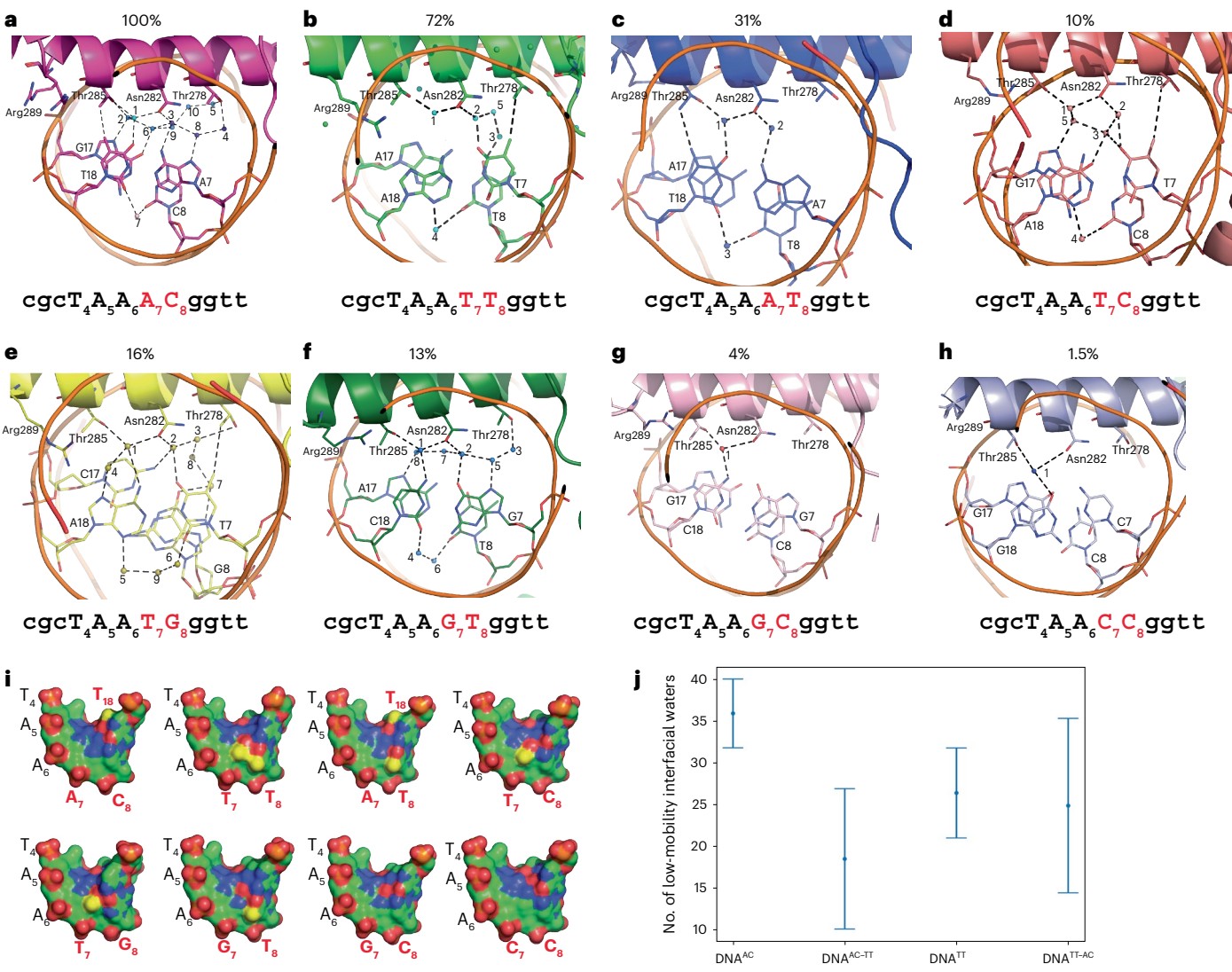

**Fig. 3 | The role of water molecules in BARHL2–DNA complexes. a–h,** The analysis of eight structures of BARHL2 containing the different dinucleotides: BARHL2–DNA$^{AC}$ (**a**), BARHL2–DNA$^{TT}$ (**b**), BARHL2–DNA$^{AT}$ (**c**), BARHL2–DNA$^{TC}$ (**d**), BARHL2–DNA$^{TG}$ (**e**), BARHL2–DNA$^{GT}$ (**f**), BARHL2–DNA$^{GC}$ (**g**) and BARHL2–DNA$^{CC}$ (**h**). The enrichment of the respective 8-mer relative to TAAACG is shown above the panels. The enrichment of a specific 8-mer relative to TAAACG is calculated as the percentage ratio of its fold change—defined as the ratio of normalized counts (count of the 8-mer divided by the total number of reads) in signal and background files—to the fold change of TAAACG. The water molecules are numbered for clarity. The hydrogen bond distances found in BARHL2–DNA complexes are presented as dashed lines and are listed in Extended Data Table 1. Panels **b**, **c**, **d**, **g** and **h** show the complexes containing the largest number of water molecules of the four complexes of the respective asymmetric units.

The structures in **a** and **f** contain only one complex in the asymmetric unit. **i,** Surface representation of the DNA-binding sites. Carbon, oxygen, nitrogen and phosphorus atoms are green, red, blue and orange, respectively and the CH3 group of thymidine is yellow. **j,** Analysis of interfacial water mobility using molecular dynamic simulation. Left, simulations run using the DNA$^{AC}$ lattice with or without in silico mutation of the AC dinucleotide to TT; right, simulations run using the DNA$^{TT}$ lattice with or without in silico mutation of the TT dinucleotide to AC. Note that in both cases, AC dinucleotide has a higher percentage of low-mobility waters at the protein–DNA interface. Data are presented as mean values; error bars, s.e.m. For DNA$^{AC}$ and DNA$^{AC-TT}$, the sample size is four (dynamically independent copies of the protein–DNA complex within the crystallographic unit cell); similarly, for DNA$^{TT}$ and DNA$^{TT-AC}$, the sample size is eight.

---

to BARHL2 than both optimal sequences. The DNA recognition pattern in the BARHL2–DNA$^{AT}$ complex is more similar to DNA$^{AC}$ than DNA$^{TT}$. Compared to DNA$^{AC}$, the hydrophobic contact between Thr285 and $T_{18}$ opposite to $A_7$ of AC is retained, but the extensive water network connecting both Thr285 and Thr278 with the AC region is almost completely lost. There is only one water molecule connecting the side-chain oxygen of Thr285 with the side-chain oxygen of Asn282 and one water molecule connecting Asn282 with N6 of $A_7$ of AC (Fig. 3c). In addition, the side chain of Arg289 has only one conformation, rotated away from the protein–DNA interface.

Another sequence, DNA$^{TC}$ is also only one Hamming distance away from both local optima but has an even weaker affinity to BARHL2 than

DNA$^{AT}$. The recognition pattern appears to be a mixture of DNA$^{TT}$ and DNA$^{AC}$, with the hydrophobic contact of DNA$^{TT}$ between $C_7$ of $T_7$ and the methyl group of Thr278 retained but at a sub-optimal distance (4.3 Å; Fig. 3d). The water molecules are arranged near Asn282 in a somewhat similar pattern to those observed in DNA$^{AC}$, with the water net connecting the side-chain oxygens of Thr285 and Asn282 and N7 and O6 of $G_{17}$ (opposite to C of AC). Thus, DNA$^{TC}$ combines aspects of recognition from DNA$^{TT}$ and DNA$^{AC}$, but both types of interactions are less optimal, explaining the relatively low affinity of the BARHL2–DNA$^{TC}$ complex.

To compare, we solved BARHL2 bound to four other sequences that are close to one but not the other optima. Mutations of the second T ($T_8$) of DNA$^{TT}$ to $G_8$ decreased affinity less than the mutation to C. Overall,

the recognition pattern in the DNA$^{TG}$ complex is similar to that observed in the DNA$^{TC}$ (Fig. 3e). However, the non-polar contact between Thr278 and T$_7$ has a more optimal distance. The water network is also more extensive, with three more water molecules. Those water molecules connect the side-chain oxygen of Thr278 to both O6 and N7 of one of the observed conformations of G$_8$. The DNA$^{TG}$ complex also shows very strong distortion of the DNA backbone, with a 3.7 Å shift between two conformations of the phosphate backbone at the position of the G nucleotide located after the TG sequence (Extended Data Fig. 3f). By contrast, the replacement of T$_7$ of DNA$^{TT}$ with G led to a substantial loss of affinity; in BARHL2–DNA$^{GT}$, neither Thr278 nor Thr285 makes direct hydrophobic contact with bases. Instead, water molecules are spread rather equally near both amino acids. The side-chain oxygen atoms of Thr285, Asn282 and N4 of C$_{18}$ are in contact through one water, which is connected to the other one contacting N7 of A$_{17}$ and simultaneously participating in a five-water chain connecting both O6 and N7 of G$_7$ with the side-chain oxygen atoms of Asn282 and Thr278 (Fig. 3f).

Mutation of the AC sequence led to a much larger loss of affinity than mutation of TT. Mutating the A$_7$ to either G or C of DNA$^{AC}$ resulted in a loss of most water-mediated contacts and a very low affinity. Both DNA$^{GC}$ and DNA$^{CC}$ complexes showed only one water molecule in the interface connecting the side-chain oxygens of Thr285 and Asn282 with the N4 of C$_{18}$ or O6 of G$_{18}$, respectively (Fig. 3g,h). Furthermore, neither complex retained the hydrophobic contacts involving either Thr278 or Thr285, further explaining their very low affinity.

### Role of hydrophobic interactions and water-mediated interactions

To determine the role of hydrophobic interactions and water-mediated bonds in BARHL2–DNA interactions, we mutated the two threonines (Thr278 and Thr285) that contribute to the hydrophobic interactions and water-chain organization to residues found in other homeodomain proteins (Extended Data Fig. 4). The protein–DNA affinities were measured using SELEX. As expected, binding of BARHL2 to the TAATT sequence, which is commonly recognized by homeodomains, was not abolished by most of the mutations (Fig. 4a). However, binding to the BARHL-family-specific TAAAC sequence was very sensitive to mutation. Significant binding to TAAAC was retained only in two cases, where Thr278 was mutated to residues that retained the methyl but not the hydroxyl group (isoleucine or valine, residues present in NKX1.2 and EMX1, respectively). These results support the importance of Thr278 and Thr285 in organizing the water network that recognizes the BARHL2-specific TAAAC sequence.

To further assess the role of hydrophobic contacts, we tested the effect of cytosine methylation. Introduction of a hydrophobic methyl group to the five position of C enables methyl-C to take part in similar hydrophobic interactions as a T. In addition, the methyl group destabilizes local water networks. To test the effect of methylation of Cs at all positions and dinucleotide contexts, we directly introduced 5-methylcytosine to DNA using PCR and then determined the enrichment of different sequences using SELEX. Two strong effects were detected: first, the methylation of C-base in DNA$^{AC}$ reduced BARHL2–DNA affinity (Fig. 4b). The lower affinity was probably a result of the methyl group interfering with the elaborate water network of the BARHL2–DNA$^{AC}$ complex (Fig. 4b). Second, methylation of the C on the complementary strand of DNA$^{GT}$ greatly increased the affinity, to a level even higher than that observed in both unmethylated optimal sequences (DNA$^{AC}$ and DNA$^{TT}$). The high affinity is probably a result of the ability of mC but not unmethylated C to form a hydrophobic contact with Thr285 (Fig. 4b–d).

To test the role of enthalpic water-mediated interactions in BARHL2–DNA binding, we performed the HT-SELEX at a series of different temperatures (0 °C, 10 °C, 20 °C, 30 °C, 40 °C). As the entropic contribution to binding depends on temperature, comparing binding at different temperatures can reveal which sequences are bound

more entropically and which depend more on enthalpic contribution to affinity. A comparison of the effect of temperature on the relative binding affinity of the different sequences revealed that the affinity to the enthalpic TAAAC sequence decreased when the temperature was increased; a similar but less dramatic trend was also observed for TAAGT. However, the sequence representing the entropic optima TAATT was less affected by temperature (Fig. 4e). Taken together, these results are consistent with the role of extensive water networks in the binding of BARHL2 to the TAAAC and methylated TAAGT sequences and a dominant entropic contribution of binding to TAATT[38].

### Molecular dynamics analysis of entropy

To assess the role of water entropy in the binding process of BARHL2 to DNA$^{AC}$ and DNA$^{TT}$, we performed molecular dynamics simulations and calculated spatially resolved solvent entropies using Per|Mut[52]. Water molecules in a 1 nm thick solvation shell around the complexes were localized by permuting water identities[52]. This procedure does not change the physical properties of the water molecules but serves to estimate the rotational and translational freedom of individual waters at a given average position around the BARHL2–DNA complexes and thus allowed us to compute their individual solvent entropies as well as correlation corrections based on a mutual information expansion approach[52].

The computed solvent entropies at given average water positions of BARHL2–DNA$^{AC}$ and BARHL2–DNA$^{TT}$ indicated that waters on both interfaces are more restrained than bulk water (118 J K$^{-1}$ mol$^{-1}$; Fig. 5a,b). The BARHL2–DNA$^{AC}$ interface involves a subset of 10–15 very low-entropy waters (50–80 J K$^{-1}$ mol$^{-1}$) as well as less restrained waters (80–100 J K$^{-1}$ mol$^{-1}$). Most of the low-entropy waters form a stable hydrogen bond network connecting the recognition helix of BARHL2 with bases at the TAAAC site (Extended Data Fig. 6a), in good agreement with the water molecules observed in the two BARHL2–DNA$^{AC}$ structures resolved at 0.95 Å and 1.3 Å. The BARHL2–DNA$^{TT}$ complex simulations also show a water-mediated hydrogen bond network at the protein–DNA interface, formed by 20–35 medium-entropy (75–90 J K$^{-1}$ mol$^{-1}$) water molecules (Extended Data Fig. 6e), in agreement with the results of the crystal lattice simulations (Fig. 3j). Consistent with the relatively high predicted mobility of these waters, the level of electron density seen in crystal structure of the BARHL2–DNA$^{TT}$ complex in this region (<1σ) appears similar to that of free solvent at the limit of resolution (1.85 Å), thus explaining why only few water molecules are seen in BARHL2–DNA$^{TT}$ crystal structure at the dinucleotide recognition site. Indeed, the number and positions of the water molecules seen in the simulations within this region (ten for BARHL2–DNA$^{AC}$ and six for BARHL2–DNA$^{TT}$, four of which have low mobility; see Extended Data Fig. 7) agree very well with the numbers seen in the crystal structures.

Computing the average solvent entropy of 118 J K$^{-1}$ mol$^{-1}$ for bulk water (based on the simulations of pure optimal point charge water) also allowed us to estimate the total change of solvent entropy at the protein–DNA interface in the BARHL2–DNA$^{AC}$ and BARHL2–DNA$^{TT}$ complexes. The estimated solvent entropy loss upon binding is considerably larger (by ~1,500 J K$^{-1}$ mol$^{-1}$) for the BARHL2–DNA$^{AC}$ complex than for the BARHL2–DNA$^{TT}$ complex.

We also considered other contributions to the entropy, such as the flexibility and conformational entropy of the BARHL2 protein and the bound DNA in the respective complexes. Both the mean backbone positions of BARHL2 and the atom position root mean square fluctuations (RMSF) are very similar for the two complexes, explaining the relatively small 34 J K$^{-1}$ mol$^{-1}$ difference in BARHL2–DNA$^{AC}$ and BARHL2–DNA$^{TT}$ protein conformational entropies (Fig. 5c). This difference can be mostly attributed to the higher conformational entropy of DNA-binding side-chain residues in the BARHL2–DNA$^{AC}$ complex. The analysis of residue-wise RMSF values for the two DNA sequences (Fig. 5d) shows a larger difference in conformational flexibility, again in favor of

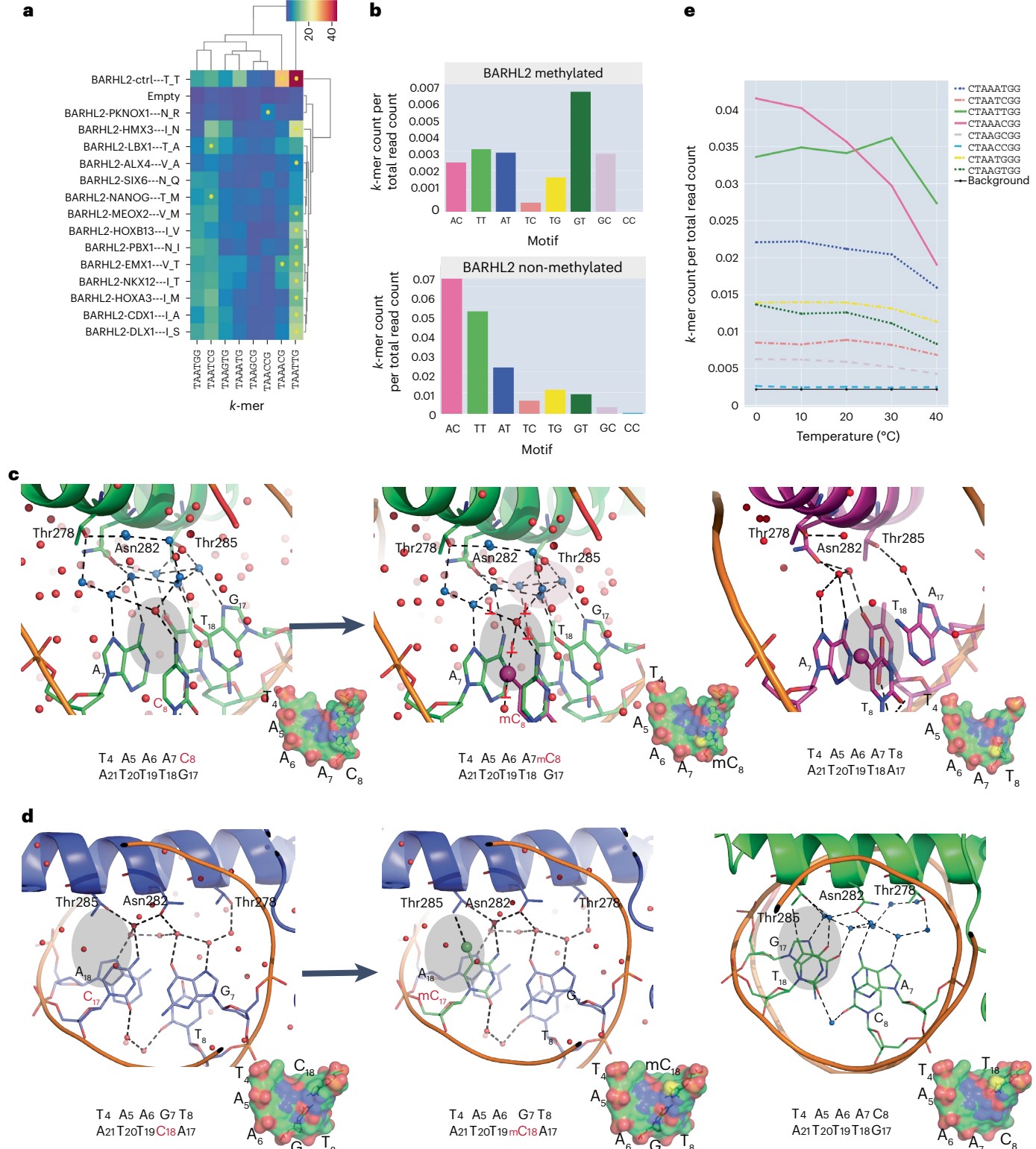

**Fig. 4 | Effects of mutations, methylation and temperature changes on BARHL2 binding to DNA. a**, The effect of mutation of Thr278 and Thr285 on the binding of BARHL2 to the different homeobox-like sequences. Color indicates relative enrichment of the indicated 6-mer in the fourth SELEX cycle compared to the initial DNA library. Yellow circles indicate that the 6-mer represents local maxima (enrichment is higher than that of related sequences). Labels indicate the amino acids into which Thr278 and Thr285 were mutated, and the TF that contains those residues. **b**, The effect of methylation of the homeobox-like sequences on affinity. Top and bottom panels show relative enrichment of TAANNG 8-mer containing the indicated dinucleotide in the NN position in the presence and absence of cytosine methylation, respectively. **c**, Effect of

methylation of the C in BARHL2–DNA$^{AC}$. The five-methyl group of $C_8$ is shown in ball-and-stick and colored magenta. Note that the addition of a methyl group to DNA$^{AC}$ (left) makes the DNA$^{AmC}$ (middle) similar to DNA$^{AT}$ (right) and destabilizes the water network between BARHL2 and DNA. **d**, BARHL2 has relatively low affinity to DNA$^{GT}$ (left); however, methylation of C complementary to the G (middle) increases the affinity, probably because similar to the T complementary to the A of DNA$^{AC}$ (right), the five-methyl group of $C_{18}$ can form a hydrophobic contact to Thr285. **e**, Effect of temperature on BARHL2 affinity to different homeobox sequences. Note that temperature affects the affinity of $k$-mers containing AC (magenta) and TG (dark green) more than those containing TT (light green).

the BARHL2–DNA$^{AC}$ complex. Here, the increased flexibility of DNA$^{AC}$ can mostly be attributed to the base pairs downstream of the binding site (bases 11–14), amounting to an entropy difference of 88 J K$^{-1}$ mol$^{-1}$.

The strongly localized and extensive water network that connects BARHL2 amino acids to the AC dinucleotide is enthalpically favorable for binding but entropically unfavorable. By contrast, the TT dinucleotide exposes a more hydrophobic surface to solvent molecules in the major groove, which leads to a more mobile water network at the complex interface that is entropically more favorable than that of the AC dinucleotide. These results are in line with the entropies obtained from the isothermal titration calorimetry (ITC) experiments[38] and support the notion that solvent entropy at the binding interface contributes markedly to the entropy difference between the BARHL2–DNA$^{AC}$ and BARHL2–DNA$^{TT}$ complexes. Note, however, that for a fully quantitative comparison, further terms would have to be included, such as the entropy of water molecules released during BARHL2 binding as well as the conformational entropies of unbound BARHL2 and free double-stranded DNA molecules.

## Discussion

Here, we used structural biology to determine how two TFs, MYF5 and BARHL2, can specifically recognize dinucleotides. Extensive prior work has established the central mechanisms by which proteins interact with DNA[4,6,7,10,53,54]. Direct interactions between individual amino acids and DNA bases can differentiate between mononucleotides; however, because of their one-to-one nature, individual contacts are generally not very sensitive to neighboring base content. The influence of sequence on DNA conformation, in turn, can be used to recognize dinucleotides. For example, geometric aspects of DNA such as the width of the minor groove are affected by dinucleotide content, and narrowing of the minor groove caused by homopolymeric stretches of A or T can be sensed by arginine residues[53]. Furthermore, the bending modulus of the DNA helix towards a particular direction is sensitive to local dinucleotide content[5,55], enabling proteins to indirectly read dinucleotides by bending DNA. However, the local effect of DNA bending on dinucleotide preference is relatively small within a short segment of DNA, such as that bound by TFs[56]. In a nucleosome, which binds much longer DNA segments, the small effects add up to allow the sequence to contribute to nucleosome positioning. However, a local bend, even as strong as what is seen in a nucleosome, would not cause dinucleotide preferences as strong as what are seen in BARHL2 and MYF5. Furthermore, the effect of a protein alpha-helix inserting into the major groove of DNA has a much smaller effect on DNA backbone shape than the formation of a nucleosome[22,57]. Therefore, a mechanism based on local DNA bending alone can not account for the dinucleotide specificity of both MYF5 and BARHL2. The results we present here strongly suggest that dinucleotide recognition critically depends on water molecules at the protein–DNA interface.

Both the free DNA and the interface between protein and DNA contain large numbers of water molecules. It is well established that DNA sequence can be read by water-mediated interactions[3,7,9,39,58], and this mechanism has been observed in a large number of protein–DNA structures[3,7] and validated experimentally by mutagenesis[59–61]. Mutational evidence also suggests that a single water molecule can contribute to epistatic interactions at the trp repressor binding site by recognition of both CT (optimal sequence) and TC (-15-fold lower affinity) dinucleotides[60]. However, to our knowledge, no previous work has solved structures of TFs bound to two sites containing distinct dinucleotides that display similar binding affinity.

Importantly, the indirect nature of the water-mediated interactions commonly leads to large tradeoffs between entropy and enthalpy[39–41]. We have shown earlier that individual TFs can bind to two distinct locally optimal sequences: one representing entropic optima, where water molecules are more mobile, and the other representing enthalpic optima, where water molecules at the protein–DNA interface are less mobile and form complex networks[38]. We show here that these water correlations contribute to the recognition of a dinucleotide within the BARHL2 recognition sequence (Fig. 5), which is recognized either enthalpically by an extensive water network that links the two DNA nucleotides to the same amino acid (BARHL2–DNA$^{AC}$) or entropically by formation of a local hydrophobic patch that repels water (BARHL2–DNA$^{TT}$). In both cases, the dinucleotides are preferred because of the local collective action of the water molecules. Sequences between TT and AC lead to compromises that weaken affinity because water molecules have a strong influence on each other in the confined space between the protein and DNA. The two modes of dinucleotide recognition also lead to a different temperature sensitivity of binding of the same TF to two different motifs. This effect may result in differential gene expression of target genes that contain the entropic and enthalpic sites, enabling organisms to directly sense their body temperature at the level of individual genes[2]. This mechanism is likely to be particularly important in unicellular organisms, plants and poikilothermic animal species.

Recently, computer simulation methods that provide spatially resolved maps of hydration thermodynamics in protein–ligand systems have been developed. Spatial decomposition of translational water–water correlation entropy in Factor FXa[62] and Crambin[52] showed that thermodynamic driving forces linked to hydration can have an important role in protein folding and ligand binding[63]. In the case of Crambin, more than half of the solvent entropy contribution came from induced water correlations[52], highlighting the importance of water–water interactions. Here, we used the Per|Mut algorithm to analyze the entropic contributions of solvent at the binding interface of the BARHL2–DNA$^{AC}$ and BARHL2–DNA$^{TT}$ complexes. Based on these solvent entropy calculations, we estimated that the solvent entropy loss is considerably larger for BARHL2–DNA$^{AC}$ than for BARHL2–DNA$^{TT}$. The difference is a significant contribution to the free-energy budget of

**Fig. 5 | Analysis of entropic and enthalpic optima using molecular dynamics. a**, Molecular dynamics analysis of entropies of resident water molecules at the BARHL2–DNA$^{AC}$ and BARHL2–DNA$^{TT}$ interface, colored according to their solvent entropy. Below, number of waters at the complex interface, their average ($S_{avg}$) and total ($S_{total}$) solvent entropy as well as the estimated solvent entropy loss ($\Delta S_{loss}$) of interfacial waters is shown for both complexes. **b**, Distribution of solvent entropies of interfacial water molecules compared to bulk solvent waters. **c**, Flexibility of BARHL2 protein main chain (solid lines) and all atoms (dotted lines) bound to DNA$^{AC}$ (magenta) and DNA$^{TT}$ (green). **d**, Flexibility of BARHL2 bound DNA$^{AC}$ (magenta) and DNA$^{TT}$ (green) DNA backbone (solid lines) and all DNA atoms (dotted lines). **e**, Partial contributions to the entropy difference of BARHL2–DNA complexes including the conformational (Schlitter) entropies for the protein and DNA as well as the entropy contribution of interfacial waters. **f,g**, Surface representation of the protein–DNA interface near the variable bases AC (**f**) and TT (**g**). Only the recognition helix from BARHL2 (wheat) is represented

for clarity. The variable bases are in yellow, and the methyl group (CH3) of T is in violet. The red and teal spheres represent the water molecules observed in the interface and minor groove, respectively. **h,i**, Schematic representation of the two optimal sites representing enthalpy (**h**) and entropy (**i**) optima. Only the recognition helices and the divergent dinucleotides are shown for clarity. The amino acid residues involved in the recognition are presented as sticks. The bases are color-coded as follows: adenine, green; thymine, red; cytosine, blue; guanine, yellow. Note that the high-entropy state (TT) has very few fixed water molecules whereas the enthalpic state (AC) contains several properly fixed water molecules that are used for the formation of the hydrogen bonds linking BARHL2 to DNA. The small panels in **h** and **i** show the distribution of the hydrogen bond partners on the surface of the optimal sequences. Positively charged atoms (nitrogen) are blue; negatively charged atoms (oxygen) are red; and carbon and phosphorus are green and orange, respectively.

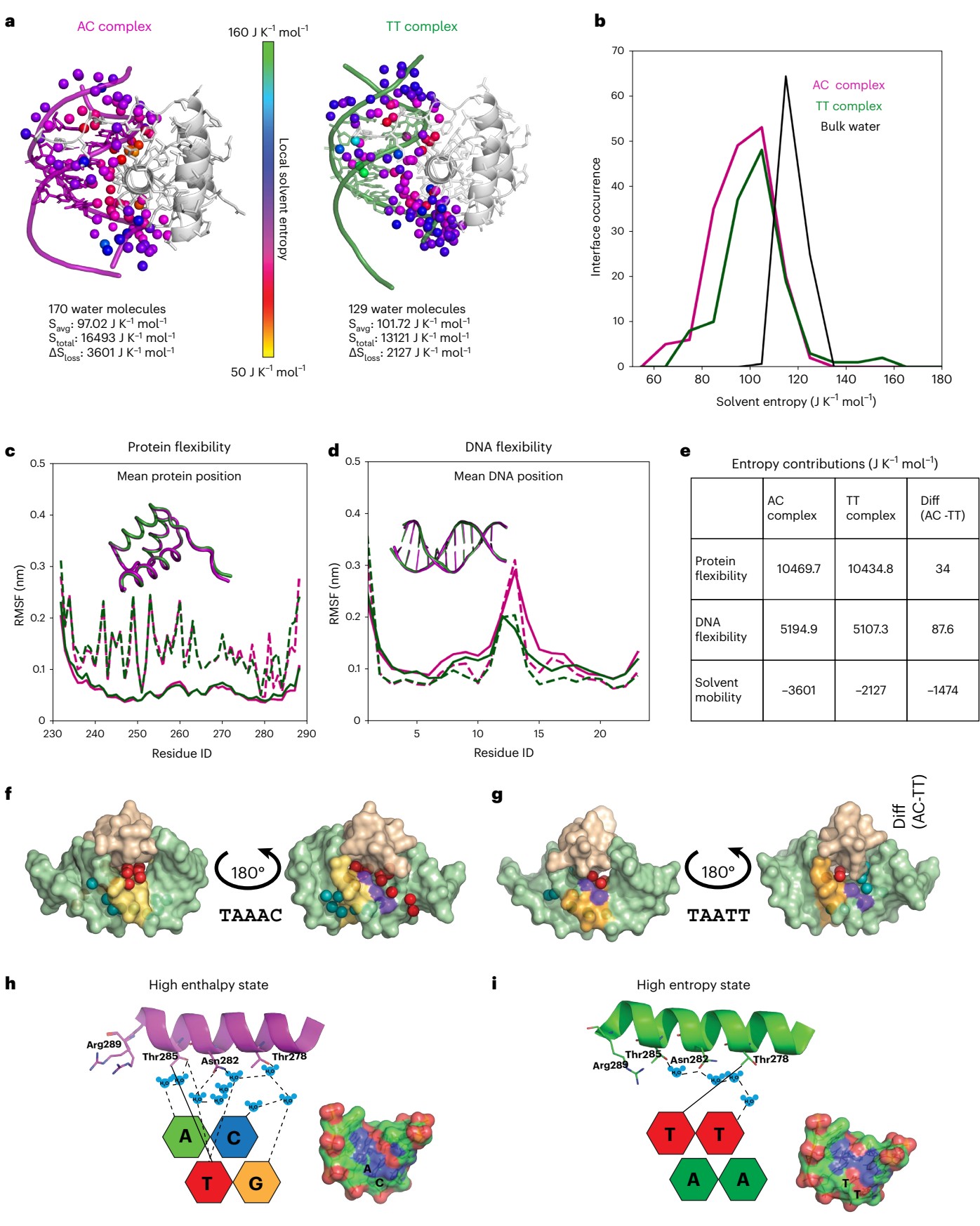

the BARHL2 binding, and probably the main source of binding entropy difference observed in the ITC measurements[38].

In summary, our work represents the largest high-resolution structural study of the effect of DNA sequence on protein–DNA interactions. The extensive analysis highlights the role of water in the molecular recognition of dinucleotides and in the recognition of features that are larger than those that can be bound by an individual chemical bond. Recent advances in the prediction of protein structures from sequence[64,65] raise the possibility that another difficult problem—determining the affinity of macromolecular interactions from sequence—could also be addressed by machine learning. However, the non-additive nature of the solvent interactions increases the complexity of the problem and highlights the need for explicitly considering both enthalpy and entropy in building computational models of macromolecular recognition. Further work combining high-throughput methods described here, including temperature SELEX, with molecular dynamics simulation-based entropy calculations are promising avenues for generating the large-scale data necessary for building computational models that seek to solve the relationship between macromolecular sequence and affinity.

## Online content

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

## Methods

### Protein expression, purification and crystallization

Expression and purification of the DNA-binding domain fragment of human MYF5 (residues 82–136) as well as BARHL2 (residues 232–292) were performed as previously described[44,66]. The DNA fragments used in crystallization were obtained as single-strand oligonucleotides (Eurofins) and annealed in 20 mM HEPES (pH 7.5) containing 150 mM NaCl and 0.5 mM Tris (2-carboxyethyl) phosphine (TCEP) and 5% glycerol. For each complex, the purified and concentrated protein was first mixed with a solution of annealed DNA duplex at a molar ratio of 1:1.2 and, after 1 h on ice, subjected to the crystallization trials. The crystallization conditions for both MYF5 complexes were optimized using an in-house-developed crystal screening kit combining different polyethylene glycols (PEGs) with different additives. Complexes of MYF5 with symmetrical DNA were crystalized in sitting drops by a vapor diffusion technique from a solution containing 50 mM sodium acetate buffer at pH 4.5, 10% PEG (3350) and 2% 2-methyl-2,4-pentanediol (MPD). Complexes of MYF5 with non-symmetrical DNA were also crystallized from 0.05 M sodium acetate buffer at pH 4.5 but containing 11% PEG (1000), 2% MPD and 5% PEG (400). All crystals of BARHL2 complexes with different DNAs were obtained in the same conditions from the reservoir solution containing 100 mM sodium acetate buffer (pH 4.8), 34% PEG (1000) and 0.06 M sodium malonate (pH 7.0) All datasets were collected at the European Synchrotron Radiation Facility (ESRF) from a single crystal on beamline ID23-1, at 100 K using the reservoir solution as cryo-protectant. Data were integrated with the program XDS[67] and scaled with SCALA[68]. Statistics of data collection are presented in Table 1.

### Structure determination and refinement

All structures were solved by molecular replacement using the program Phaser 2.8.3 as implemented in Phenix 1.20.1-4487[69] and CCP4 suits 7.1 and 8.0 (ref. 70) the structure of MYOD (PDB 1MDY) as a search model for MYF5 and the structure of *Drosophila* clawless homeodomain protein (PDB 3A01, chain A) as a search model for BARHL2. After the positioning of the protein, the density of DNA was clear and the molecule was built manually using COOT 0.9.6[71]. The rigid body refinement with REFMAC5 5.8.0267 was followed by restrain refinement with REFMAC5, as implemented in CCP4 (ref. 70) and Phenix.refine[72]. The manual rebuilding of the model was done using COOT. The refinement statistics are presented in Table 1. In both MYF5 complexes, all residues as well as all DNA bases were well defined in the electron density maps. In the structure of MYF5$^{GT/AC}$, only three water molecules were well defined, whereas in the structure of MYF5$^{AA/AC}$, 71 water molecules were traced. All residues and bases on all BARHL2 complex structures were well visible in the maps. Figures showing structural representations were prepared using PyMOL 2.5.4[73].

### Molecular dynamics simulations in crystal lattice

The program Coot[71] was used to in-silico mutate the TT dinucleotide to AC and vice versa at every monomer of the asymmetric unit and to energy minimize the resulting mutant structures before building the lattice for the simulations. The positions of the BARHL2 crystallographic water molecule oxygens were not changed from those of the initial crystal structure.

Four crystal lattice simulations (BARHL2–DNA$^{AC}$, BARHL2–DNA$^{AC→TT}$, BARHL2–DNA$^{TT}$, BARHL2–DNA$^{TT→AC}$) were then performed using AMBER21 (ref. 74); models of the complete unit cell were created using UCSF Chimera 1.0[75], neutralizing sodium ions added using the AMBER *AddToBox* tool, and then the same tool was used to add sufficient additional waters to achieve a final unit cell density of approximately 1.0. Control simulations of the DNA alone in water were prepared from the crystal structure coordinates using the Amber *tleap* tool, adding neutralizing sodium ions and enough water to fill a truncated octahedral periodic box extending a minimum of 10 Å beyond any solute atom. Systems were parameterized using the FF14SB force field[76] for the protein component, with additional BSC1 parameters for the DNA[77]. Waters and ions were modeled using the TIP3P and Joung–Cheatham[78] parameters, respectively.

Crystal lattice simulations were performed using *pmemd.cuda*. Unless specified otherwise, all simulations used default parameters. Systems were first energy-minimized with restraints (0.1 kcal mol$^{-1}$ Å$^{-2}$) on all atoms except modeled waters (that is, waters observed in the crystal structures were in the restrained group) before a second energy-minimization step without restraints. Molecular dynamics simulations used Langevin dynamics with a collision frequency of 5 ps$^{-1}$ and a 2 fs timestep, with SHAKE on all bonds to hydrogen atoms. Non-bonded interactions were handled using the Particle Mesh Ewald method, with a direct space cutoff of 9 Å. Snapshots were saved every 100 ps. For the crystal lattice simulations, molecular dynamics began with a 10 ns NVT simulation, with a target temperature of 300 K and restraints of 0.01 kcal mol$^{-1}$ Å$^{-2}$ on all solute atoms, followed by a 50 ns NVT simulation in which only DNA atoms were restrained (same force constant). For the 50 ns production NVT simulation, only DNA C1′ atoms remained restrained. For reference simulations of the DNA alone in solution, molecular dynamics began with a 50 ns NPT simulation at 300 K (pressure coupling parameter 2 ps$^{-1}$) with restraints (0.01 kcal mol$^{-1}$ Å$^{-2}$), followed by a 200 ns NPT production simulation in which the only restraints were on the Watson–Crick hydrogen bonds in the terminal base pairs (a flat-bottomed potential that was zero between 1.9 Å and 2.1 Å and had a force constant of 10 kcal mol$^{-1}$ Å$^{-2}$ outside this range).

DNA helical parameters were analyzed using the AMBER21 cpptraj tool[79]. Other analyses and visualizations were performed in Jupyter notebooks, using the MDTraj[80] and Matplotlib packages. The method to analyze water molecule mobility involves first permuting the indices of the water oxygen atoms in each frame in the trajectory so that each remains within a small region of space. The mean position and fluctuation of each water can then be measured. The re-indexing is an iterative process using the linear sum assignment approach; the Python code to implement this procedure is available at http://github.com/CharlieLaughton/water_shuffel.

To determine the number of low-mobility water molecules at the protein–DNA interface, the RMSF of all interfacial waters was calculated. A water molecule was considered interfacial if its oxygen atom was simultaneously less than 5 Å of a protein atom and a DNA atom. Interfacial water molecules were considered of low mobility if the RMSF of their oxygen atom was smaller than 1.9 Å.

To check equilibration and sampling, this approach was applied independently to data for the first and second halves of the production phase of each simulation. Results are shown in Extended Data Fig. 5. There is no sign of systematic change in values between the first and second subsamples, and all trends between different molecular systems are maintained.

### Molecular dynamics simulations for analysis of entropy

To determine the entropic contributions of DNA, protein and water molecules, we performed all-atom molecular dynamics simulations of two BARHL2–DNA complexes with the sequences CGCTAAACGGTT (AC complex) and CGCTAATTGCTC (TT complex) in aqueous solution using the GROMACS 2019 simulation package[81]. Simulations of both complexes were performed in four replicates, starting from conformations derived from still unpublished crystallographic structures of the complexes, including crystallographic waters around the protein–DNA complexes. Each replicate trajectory was propagated using a leapfrog algorithm[82] in 2 fs timesteps for 5 μs, resulting in a total 20 μs simulation time for each complex.

All simulations were kept at 298.15 K using a velocity rescale thermostat[83] and a coupling constant of 0.1 ps. The pressure during simulations was maintained at 101 kPa using an isotropic Parrinello–Rahman

barostat[84], an isothermal compressibility of $4.5 \times 10^{-7}$ kPa$^{-1}$ and a coupling constant of 30 ps. The intramolecular and intermolecular interactions of the protein and DNA were described by the AMBER-19SB force field[85]. All simulations were performed under periodic boundary conditions in a cubic simulation box of 7.8 nm edge length filled with optimal point charge waters[86], as well as Na$^+$ and Cl$^-$ ions appropriate for a 300 mM sodium chloride salt solution to match the experimental conditions of the ITC experiments. Electrostatic and van der Waals interactions were explicitly calculated within a 1.0 nm cutoff distance; long-range electrostatic interactions beyond this cutoff were calculated by Particle Mesh Ewald summation[87] with a grid spacing of 0.13 nm. Long-range van der Waals dispersion corrections[88] to the total energy of the system were applied in all simulations. Fast vibrational degrees of freedom were removed by using the LINCS algorithm[89] as implemented in GROMACS 2019 using a fourth-order iterative restraint on the bond angles.

For this calculation, four ×500 conformations per BARHL2–DNA complex were taken from molecular dynamics simulation trajectories at 10 ns intervals. Local protein and DNA flexibility was analyzed by calculating the atom position RMSF for each solute atom using the gmx rmsf program of the GROMACS 2019 simulation package[81] and computing residue-averaged RMSF values for protein amino acids and DNA nucleic acids. Solute entropies were estimated from the same conformations by first computing the atom position covariance matrix using the gmx covar program. The maximum solute entropy of both protein and DNA was estimated from the covariance matrix using Schlitter's entropy formula[90] as implemented in the gmx anaeig program of GROMACS 2019 (ref. [81]).

Entropies of the solvent shell around the BARHL2–DNA complexes were computed from the simulation trajectories using the Per|Mut software[91]. In brief, 2,490 water molecules nearest to the protein–DNA complex (approximately a water shell of 1 nm thickness) were extracted from the molecular dynamics trajectories and subjected to permutation reduction. This procedure permutes the labels of the water molecules such that these always remain closest to their initial reference position without changing the underlying physics, thereby vastly reducing the amount of sampling required to converge solvent entropy calculations for N water molecules by the Gibbs factor (N!). The permuted water trajectories were used to compute solvent entropy estimates for particular water positions around the BARHL2 complexes, including first-order and second-order correlation terms for rotational and translational degrees of freedom of pairs of water molecules. We used the first-order translational entropy of waters to estimate their mobility at given positions, which is expected to determine their visibility in crystallographic structures.

The computed solvent entropies of individual water positions were mapped to the mean position of that given water around the BARHL2–DNA complex and analyzed further to determine the solvent entropy at the complex interface. Water molecules were identified as interfacial waters of the BARHL2–DNA complex if their mean position was simultaneously within 4 Å distance of both a protein atom and a DNA atom of the reference structure. Selection of the interfacial waters was performed with the software PyMol (v.2.5.4)[73], using the initial (crystallographic) structure as a reference.

The change of solvent entropy at the BARHL2–DNA complex interface $\Delta S_{solv}$ relative to bulk was computed according to:

$$\Delta S_{solv} = N\left(\langle S_{solv}^{int}\rangle - \langle S_{solv}^{bulk}\rangle\right),$$

where $N$ is the number of water molecules at the complex interface, and $\langle S_{solv}^{int}\rangle$ and $\langle S_{solv}^{bulk}\rangle$ = 118 J K$^{-1}$ mol$^{-1}$ are the mean solvent entropies of water molecules at the interface and in bulk water, respectively. The latter was computed using a separate simulation of bulk water using the same simulation parameters as used for the protein–DNA complexes, where Per|Mut was applied to a 1 nm sphere of bulk water in the middle of the box to obtain the mean solvent entropies.

Water bridges between the BARHL2 and its bound DNA were identified and characterized using the water-bridge analysis module of the Python[92] MDAnalysis package[93]. This analysis identified hydrogen bond networks between hydrogen donors and acceptors of BARHL2 and its bound DNA, either by direct hydrogen bonding (0$^{th}$-order water bridges) or by one or two water molecules (1$^{st}$-order, 2$^{nd}$-order and 3$^{rd}$-order water bridges). In brief, possible hydrogen bond acceptors (A) included all nitrogen and oxygen atoms of the protein, DNA or solvent with less than four covalent bonds; hydrogen bond donors (D) additionally required a chemically bound hydrogen (H) atom; a hydrogen bond between a donor, hydrogen and acceptor was recognized if the D–A distance was smaller than 3 Å and the D–H–A angle was larger than 120°.

The hydrogen bond network analysis on the molecular dynamics trajectories was restricted to stable intermolecular hydrogen bond networks between BARHL2 amino acids and nucleotides of its DNA-binding partner. The occurrence of intermolecular hydrogen bond networks (including specific waters) was averaged over the simulation trajectories and summed by amino acid nucleotide pair. Intermolecular hydrogen bonding between a residue pair was considered stable if the summed occurrence of related hydrogen bond networks was observed in more than 5% of all molecular dynamics conformations of the complex.

## Mutational analysis and HT-SELEX

The pETG20A_SPB vectors with BARHL2 and its mutant sequences (GenScript) were expressed in Rosetta(DE3)pLysS *E. coli* strain (Millipore). In brief, the bacteria were grown overnight at 37 °C in LB Broth medium (Gibco) with carbenicillin (0.1 mg ml$^{-1}$) and chloramphenicol (34 µg ml$^{-1}$) and then transferred to the induction medium consisting of the previous medium with additional reagents at the following final concentrations: 1 mM MgSO$_4$, metal mixture (50 µM FeCl$_3$, 20 µM CaCl$_3$, 10 µM each of MgCl$_2$ and ZnSO$_4$, 2 µM each of CoCl$_2$, CuCl$_2$, NiCl$_2$, Na$_2$MoO$_4$, Na$_2$SeO$_3$ and H$_3$BO$_3$), 60 µM HCl, NPS (50 mM KH$_2$PO$_4$, 50 mM Na$_2$HPO$_4$, 25 mM (NH$_4$)$_2$SO$_4$) and 5052 (0.5% glycerol, 0.05% glucose and 0.2% alpha-lactose)[94]. The cells were then incubated for 8 h at 37 °C, followed by 40 h at 17 °C. The cells were then collected by centrifugation and lysed by shaking for 20 min in 100 µl of lysis buffer (0.5 mg ml$^{-1}$ lysozyme, 1 mM phenylmethyl sulfonyl fluoride (PMSF), 400 mM NaCl, 100 mM KCl, 10% glycerol, 0.5% Triton X-100, 10 mM imidazole in 50 mM potassium phosphate buffer pH 7.8) and stored overnight at −80 °C. After thawing, the lysate was incubated for 45 min with 20 µl of Ni-Sepharose 6 Fast Flow beads (GE Healthcare), washed and resuspended in 100 µl of buffer A (30 mM NaCl, 10 mM imidazole in 50 mM Tris-HCl pH 7.5). For digestion of bacterial DNA, DNAse I and MgSO$_4$ were added to 10 µg ml$^{-1}$ and 15 mM final concentration, respectively. After incubation for 45 min at 22 °C (room temperature), the beads were washed two times with 600 µl of buffer A and two times with 600 µl of buffer A containing 50 mM imidazole. The proteins were eluted in 100 µl of buffer A with 500 mM imidazole. Protein concentration was then measured using a Bradford assay (Sigma-Aldrich, no. B6916), after which the proteins were diluted in Promega buffer (50 mM NaCl, 10 mM MgCl$_2$, 4% glycerol in 10 mM Tris-HCl pH 7.5) to 30–100 µg ml$^{-1}$ concentration.

SELEX experiments were performed as previously described[95]. In brief, 100–250 ng of proteins were mixed with 200–500 ng of DNA ligands and incubated at room temperature for 20 min; the incubation buffer contained 51.4 mM NaCl, 1.4 mM MgCl$_2$, 4% glycerol, 100 µM EGTA, 0.7 mM 1,4-dithiothreitol, 3.7 µg ml$^{-1}$ poly-dI-dC, 1.4 µM ZnSO$_4$ in 10.4 mM Tris-HCl pH 7.5 (SELEX buffer). Magnetic Ni-Sepharose beads (GE Healthcare) were washed in Promega buffer containing 0.2% BSA and suspended in 25 µl of SELEX buffer. For each protein–DNA mixture, 1.75 µl of the bead suspension was added, followed by incubation for 40 min at room temperature. The beads were then washed in wash buffer (5 mM EDTA, 1 mM 1,4-dithiothreitol in 5 mM

Tris-HCl pH 7.5) using a Tecan HydroSpeed plate washer, followed by suspension of the beads to 35 µl of elution buffer (1 mM MgCl₂, 0.1% Tween-20 in 10 mM Tris-HCL pH 7.8) followed by incubation for 10 min at 80 °C. To ensure that the ligands were double-stranded, the eluted DNA ligands were amplified twice. First, the products were amplified to completion (33 PCR cycles for the first SELEX cycle and 26 for all subsequent cycles). Then, the PCR products were diluted tenfold and amplified for two PCR cycles to ensure that single-stranded DNA and annealed products containing mismatched random sequences were fully converted to double-stranded form. The PCR reaction was performed with Phusion Hot Start II DNA Polymerase (Thermo Scientific). The DNA concentration was controlled by qPCR with 1× SYBR Green I Nucleic Acid Gel Stain (Invitrogen) using LightCycler 480 Instrument II (Roche). Amplified DNA ligands were used for incubation with proteins in the next SELEX cycle. A total of four SELEX cycles were performed. The DNA ligands from the 0, 3rd and 4th SELEX cycles were sequenced (Illumina HiSeq 2000) and analyzed as previously described[43]. For the in silico mutagenesis described above, the program Coot was used to methylate in silico the cytosine bases present in the corresponding solved structures for Fig. 4c,d.

### Motif and *k*-mer analysis

For Figs. 1 and 2, the position weight matrix models were generated from cycle four of MYF6 HT-SELEX (from ref. 18) and cycle four of the new BARHL2 (unmutated ctrl in Fig. 4a) HT-SELEX reads, using the multinomial (setting = 1) method75 with the following seeds: MYF6 AA… AC flank: AACAGCTGAC and GT…AC flank: GTCAGCTGAC; BARHL2 TT: NTAATTGN and AC: NTAAACGN. *k*-mer counts were generated using spacek40 (https://github.com/jttoivon/moder2/blob/master/myspacek40.c) for BARHL2 methylated and non-methylated SELEX data.

### Temperature HT-SELEX

To examine how the DNA-binding specificity of BARHL2 changes with temperature, HT-SELEX of BARHL2 was performed under five different temperatures 0 °C, 10 °C, 20 °C, 30 °C and 40 °C. For Temperature HT-SELEX, the DNA ligands were designed according to Illumina's Truseq library (Supplementary Table 2; 101N SELEX Ligand) and synthesized from IDT as Ultramer DNA oligonucleotides. The oligonucleotides contain a 101 bp region with randomized nucleotides, flanked by adapters of fixed sequences for amplification. First, double-stranded ligands (the input of SELEX) were synthesized from the oligonucleotides by PCR amplification with primers that match the adapters (Supplementary Table 2; PCR primers). The PCR primers were also used to amplify the library between SELEX cycles. Before sequencing, the ligands were further amplified with primers (Supplementary Table 2; PE primers) containing multiplexing indices and sequences of the Illumina flow cell (P5 or P7).

HT-SELEX was performed in microplates according to the previous protocol[18,96]. First, 100–200 ng double-stranded DNA ligand was mixed with 20–200 ng purified His-tagged TFs in 20 µl volume of the incubation buffer (140 mM KCl, 5 mM NaCl, 2 mM MgSO₄, 3 µM ZnSO₄, 100 µM EGTA, 1 mM K₂HPO₄, 20 mM HEPES pH 7.0). The mixture was incubated for 20 min on a PCR incubator (Bio-Rad S1000 Thermal Cycler) for temperature control. Then, 1.8 µl of magnetic Ni-Sepharose beads (28−9799−17, GE Healthcare; pre-blocked with 25 mM Tris-HCL, 0.5% BSA, 0.1% Tween-20, 0.02% NaN₃) was added into the mixture to pull down the TFs and their associated DNA ligands. After mixing at 1,900 rpm with a microplate shaker (13500-890, VWR), the plates were washed 15 times on a microplate washer (Tecan Hydrospeed) with the washing buffer (10 mM EDTA in 5 mM Tris-HCl pH 7.5, kept under 4 °C before use). Suspension of the washed beads was then PCR-amplified to produce DNA ligands for the next SELEX cycle. After repeating SELEX for four cycles, the ligands from each cycle and the input were sequenced and analyzed.

### Reporting summary

Further information on research design is available in the Nature Portfolio Reporting Summary linked to this article.

### Data availability

The atomic coordinates and diffraction data have been deposited to the Protein Data Bank (PDB) with accession codes PDB 7Z5I and PDB 7Z5K for MYF5 bound to symmetrical DNA and MYF5 bound to non-symmetrical DNA, respectively; PDB 8PMF for BARHL2−DNAᴬᶜ at the resolution of 0.95 Å and PDB 8PMN for BARHL2−DNAᴬᶜ at the resolution of 1.3 Å; and PDB 8PMC, PDB 8PM5, PDB 8PM7, PDB 8PMV, PDB 8PN4, PDB 8PNA and PDB 8PNC for BARHL2−DNAᵀᵀ, BARHL2−DNAᴬᵀ, BARHL2−DNAᵀᶜ, BARHL2−DNAᵀᴬᴬᴳᶜ, BARHL2−DNAᶜᶜ, BARHL2−DNAᵀᴳ and BARHL2−DNAᴳᵀ, respectively. All sequence reads have been deposited to the European Nucleotide Archive under accession number PRJEB65950. Source data are provided with this paper.

### Code availability

The Python code used for implementing re-indexing as an iterative process using the linear summation method is available at https://github.com/CharlieLaughton/water_shuffle.

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

## Acknowledgements

We thank I. Sur, C. Rogerson and B. Luisi for the critical review of the manuscript, L. Malinina for the fruitful discussion of the DNA structure at high resolution, P. Das and the Karolinska Institutet Protein Science Facility for protein production and M. Kling Pilström for technical assistance. This work was supported by the Swedish Research Council (D0815201), BBSRC/UKRI (G107673), Cancer Research UK (RG99643) and Medical Research Council (G105296). The molecular dynamics analysis of entropy was conducted within the Max Planck School Matter to Life, supported by the German Federal Ministry of Education and Research (BMBF) in collaboration with the Max Planck Society. We apologize to the authors of other structural studies of protein–DNA complexes that we unfortunately can not cite because of a limit on the number of references.

## Author contributions

E.M. performed data curation, formal analysis, investigation, visualization, writing of the original draft and reviewing and editing the final paper. Y.Y., F.Z., T.X., I.S., S.P.N., G.N. and H.G. curated the data and performed formal analysis. A.P. and C.L. performed data curation. J.T. conceptualized the project and was responsible for data curation, supervision, project administration and review and editing of the paper.

## Funding

## Competing interests

The authors declare no competing interests.

## Additional information

**Extended data** is available for this paper at https://doi.org/10.1038/s41594-024-01449-6.

**Correspondence and requests for materials** should be addressed to Jussi Taipale.

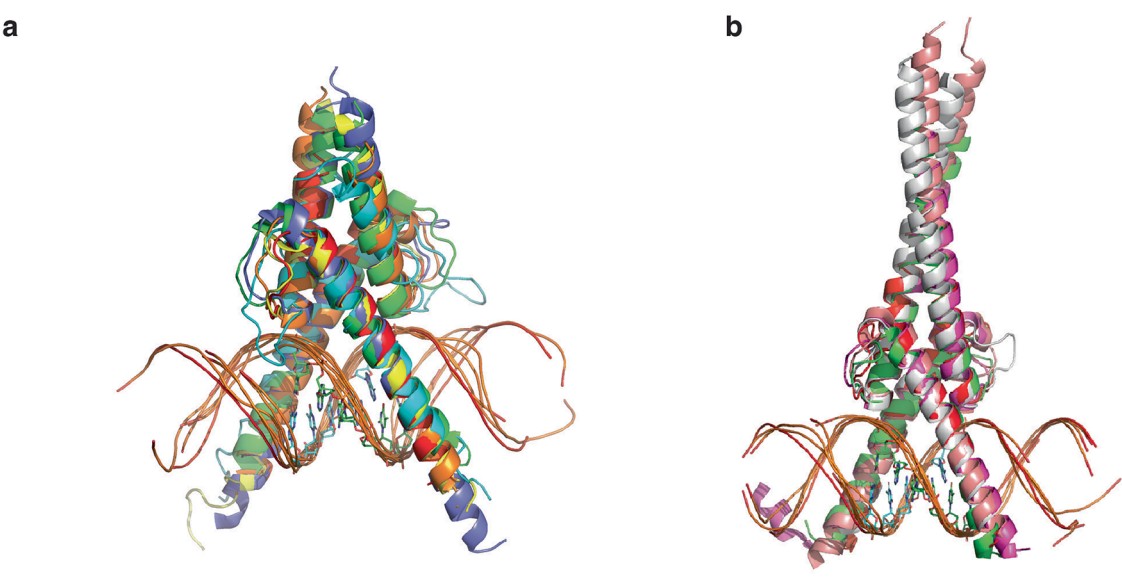

MYF5/USF/MYOD/SCL/E47/CLOCK/BMAL1/TCF4

MYF5/SREBP/MAX/MYC/MAD

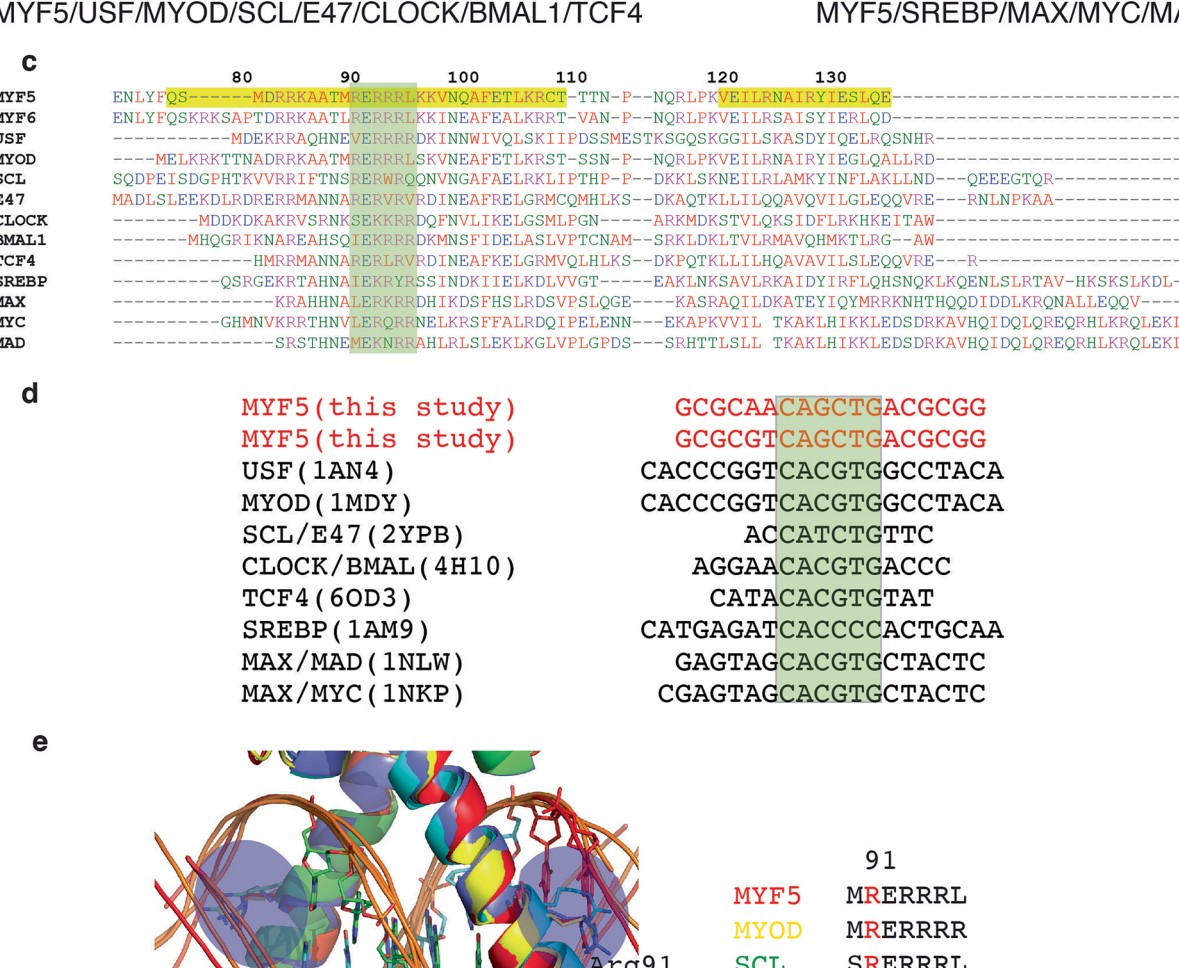

Extended Data Fig. 1 | See next page for caption.

**Extended Data Fig. 1 | Comparison of the structure of MYF5 solved in this study with different representatives of bHLH (basic helix-loop-helix) and bZIP (basic helix-loop-helix-zipper) families. a**, Structural alignment of DBD of MYF5 homodimer (red) with USF homodimer (cyan, 1AN4.pdb), MYOD homodimer (yellow, 1MDY.pdb), SCL–E47 complex (light blue, 2YPB.pdb), CLOCK–BMAL1 complex (orange, 4H10.pdb) and TCF4 homodimer (green, 6OD3.pdb). **b**, Structural alignment of MYF5 homodimer with SREBP1 homodimer (light green, 1AM9.pdb), MAX homodimer (magenta, 1HLO.pdb), MYC–MAX complex (salmon, 1NKP.pdb) and MAD/MAX complex (gray, 1NLW.pdb). The proteins used for structural alignments in **a** and **b** are listed under the figure. The DNA bases involved in the binding are presented as sticks. **c**, Sequence alignment of the DBD of proteins, used for the structural alignment in **a** and **b** with MYF6 in addition. The numbering on the top corresponds to MYF5

numbering. The helixes observed in the MYF5 structure are highlighted in light yellow. The light green box underlines the binding site. The color code for the amino acids is kept according to Clustal Omega server used for the sequence alignment (https://www.ebi.ac.uk/Tools/msa/clustalo/). Hydrophobic amino acids are colored red, polar – green, negatively charged - blue and positively charged – magenta. **d**, The DNA sequences used in the structural studies. The binding sites are highlighted with the light green box. **e**, Close view of the alignment of MYF5/MYOD/SCL/E47/TCF4 structures which are known to recognize the flanking dinucleotides. The sequence alignment shows that those bHLH TFs contain arginine residue (Arg91 in MYF5) in the specific position which is responsible for that recognition, conversely the MAX/MYC/MAD/CLOCK/BMAL1/USF/SREBP structures and sequences where arginine is replaced by different residues which are not suitable for the flank recognition.

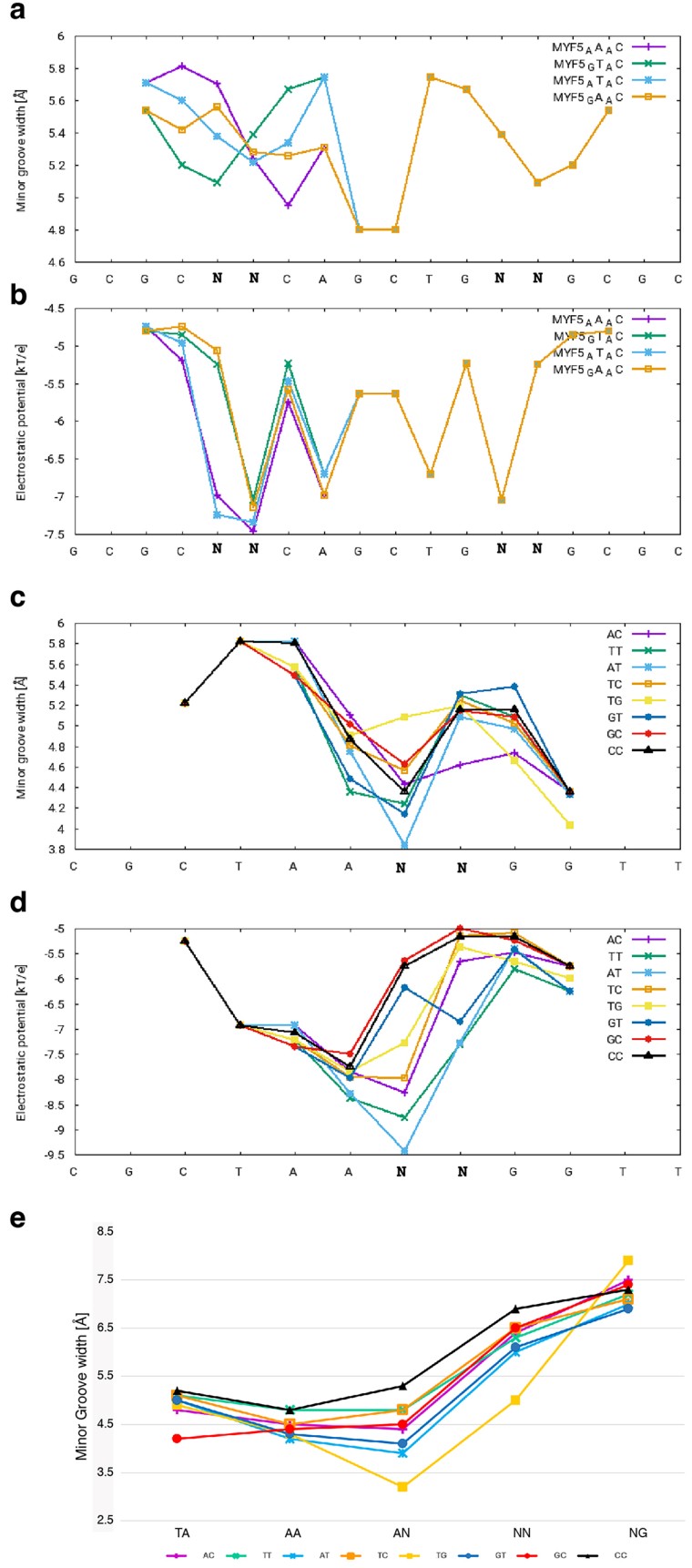

**Extended Data Fig. 2 | See next page for caption.**

**Extended Data Fig. 2 | Results of the prediction of minor groove width and electrostatic potential with *DNAphi* server (https://rohslab.usc.edu/DNAphi/serverBackend.php). a** and **b**, the predictions made for the four DNA sequences included in the study interacting with MYF5 DNA-binding domain. The violet line corresponds to the GCGC**AA**CAGCTG**AC**GCGC sequence (MYF5_aAaC), the green line corresponds to GCGC**GT**CAGCTG**AC**GCGC sequence (MYF5_gTaC); the blue line corresponds to the GCGC**AT**CAGCTG**AC**GCGC sequence (MYF5_aTaC), and the yellow line corresponds to GCGC**GA**CAGCTG**AC**GCGC sequence (MYF5_gAaC). The different nucleotides are labeled as **N** on the X-axis of the figure. **c**, **d**, the predictions made for the 8 different DNA sequences included in the study of BARHL2 DBD bound to different DNAs. Different nucleotides are labeled as **N**. The lines corresponding to different sequences are colored differently. The color legend is presented in the upper right corner and labeled by the nucleotide sequences. **e**, the minor groove width calculated with w3DNA 2.0 server (http://web.x3dna.org/) for the BARHL2 structures solved in this study. The color legend is presented under X-axis.

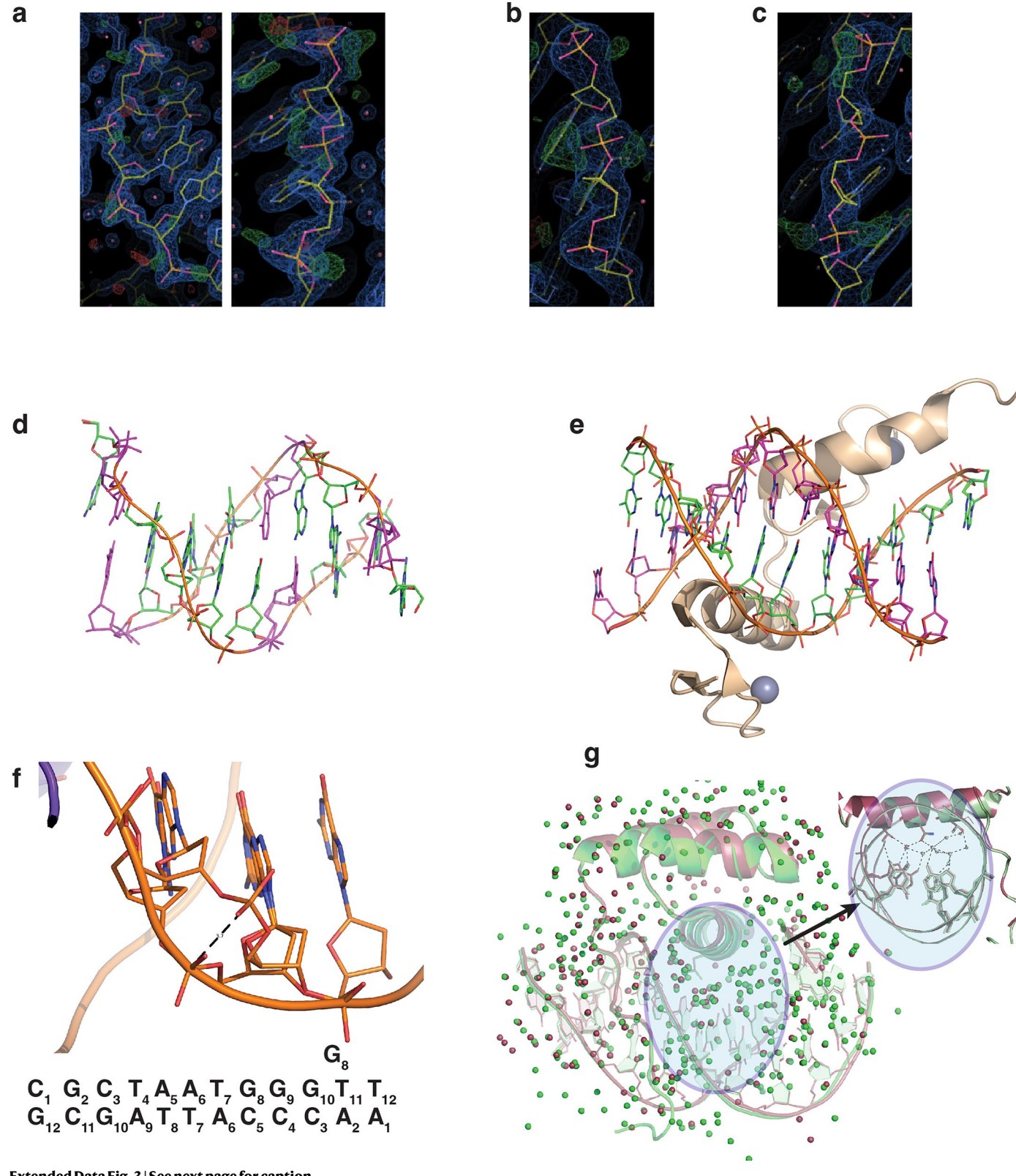

**Extended Data Fig. 3 | See next page for caption.**

**Extended Data Fig. 3 | High resolution structures of BARHL2 showed double conformations of the DNA. a**, Electron density map of the backbone phosphates in the structure with DNA TAAAAC at 1.3 Å resolution. **b, c**, Electron density map observed for the BARHL2–DNA$^{GC}$ ($G_{12}C_{11}G_{10}$) and BARHL2–DNA$^{CC}$ ($G_5C_4C_3$), respectively. **d**, The structure of the nonamer (GCGAATTCG) solved at 0.89 Å resolution showed the similar double conformation feature of the backbone phosphates (Soler-Lopez et al., 2000). The figure is created from the coordinates of 1ENN.pdb file. The backbone phosphates observed in the double conformations are colored in magenta. **e**, An example of the alternative conformation observed in the structure of protein-DNA complex. The figure is created from the coordinates of mouse Znf57-DNA complex obtained from 4GZN. pdb file. The protein moiety is colored in beige, two Zn atoms are presented in gray balls, the different DNA strands are colored in magenta and orange. Note that the most phosphates on the top strand show double conformation (Liu et.al., 2012). **f**, Double conformation of phosphates for $G_8G_9$ observed in complex BARHL2–DNA$^{TG}$. The distance between two phosphorous atoms is 3.7 Å. **g**, Structural alignment of two structures of BARHL2–DNA$^{AC}$ at the resolutions 1.3 Å (raspberry) and 0.95 Å (green). The water molecules represented as small spheres and coloured respectively to the complex. Note that the water molecules in the interface are very well conserved, particularly all 10 waters are conserved in the protein:DNA interface at the site of the variant dinucleotide, small light blue ring presents the position of the variant dinucleotides in the structure and the anel on the top right corner shows the close view of 10 conserved water molecules.

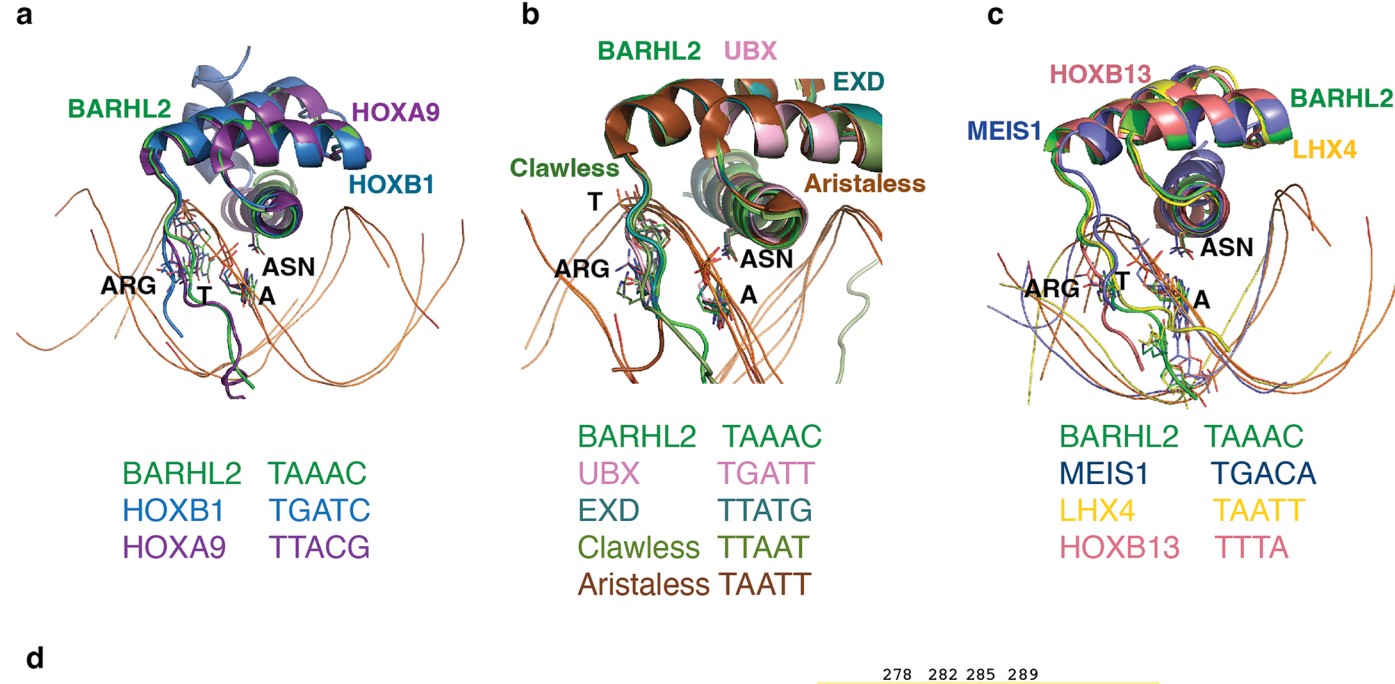

**d**

```
                                                        278  282 285  289
BARHL2    RAKKPRKARTAFSDHQLNQLERSFERQKYLSVQDRMDLAAALNLTDTQVKTWYQNRRTKWKRQTAVGLELLAE
HOXB1     GLGSPSGLRTNFTTRQLTELEKEFHFNKYLSRARRVEIAATLELNETQVKIWFQNRRMKQKKREREGG-----
HOXA9     HARSTRKKRCPYTKHQTLELEKEFLFNMYLTRDRRYEVARLLNLTERQVKIWFQNRRMKMKKINKD-------

BARHL2    RAKKPRKARTAFSDHQLNQL---ERSFERQKYLSVQDRMDLAAALNLTDTQVKTWYQNRRTKWKRQTAVGLELLAE
UBX       TNGLRRRGRQTYTRYQTLEL--EKEFHTNHYLTRRRRIEMAHALSLTERQIKIWFQNRRMKLKKEIQA-------
EDX       ----ARRKRRNFSKQASEILNEYFYSHLSNPYPSEEAKEELARKCGITVSQVSNWFGNKRIRYKKNI--------
Clawless  -MRTPPKRKKPRTSFTRIQVAELEKRFHKQYLASAERAALARGLKMTDAQVKTWFQNRRTKWRRQTAEEREAERQ
Aristaless --APKRKQRRYRTTFTSFQLEELEKAFSRTHYPDVFTREELAMKIGLTEARIQVWFQNRRAKWRKQEKV-------

BARHL2    RAKKPRKARTAFSDHQLNQLERSFERQKYLSVQDRMDLAAALNLTDTQVKTWYQNRRTKWKRQTAVGLELLAE
MEIS1     --AFPKVATNI-----MRAWLFQHLTHPYPSEEQKKQLAQDTGLTILQVNNWFINARRRIVQPM---------
LHX4      GA---KRPRTTITAKQLETLKNAYKNSPKPARHVREQLSSETGLDMRVVQVWFQNRRAKEKRLK---------
HOXB13    -----RKKRIPYSKGQLRELEREYAANKFITKDKRRKISAATSLSERQITIWFQNRRVKEKKVLAKVK-----
```

**e**

```
          223                                          278 282285       300
BARHL2    ESPPVRAKKPRKARTAFSDHQLNQLERSFERQKYLSVQDRMDLAAALNLTDTQVK TWYQNRRT KWKRQTAVGLELLAE
1. HOXB13  ESPPVRAKKPRKARTAFSDHQLNQLERSFERQKYLSVQDRMDLAAALNLTDTQVK IWYQNRRV KWKRQTAVGLELLAE
2. HOXA3   ESPPVRAKKPRKARTAFSDHQLNQLERSFERQKYLSVQDRMDLAAALNLTDTQVK IWYQNRRM KWKRQYAVGLELLAE
3. HMX3    ESPPVRAKKPRKARTAFSDHQLNQLERSFERQKYLSVQDRMDLAAALNLTDTQVK IWYQNRRN KWKRQYAVGLELLAE
4. MEOX2   ESPPVRAKKPRKARTAFSDHQLNQLERSFERQKYLSVQDRMDLAAALNLTDTQVK VWYQNRRM KWKRQTAVGLELLAE
5. ALX4    ESPPVRAKKPRKARTAFSDHQLNQLERSFERQKYLSVQDRMDLAAALNLTDTQVK VWYQNRRA KWKRQTAVGLELLAE
6. NKX12   ESPPVRAKKPRKARTAFSDHQLNQLERSFERQKYLSVQDRMDLAAALNLTDTQVK IWYQNRRT KWKRQTAVGLELLAE
7. DLX1    ESPPVRAKKPRKARTAFSDHQLNQLERSFERQKYLSVQDRMDLAAALNLTDTQVK IWYQNRRS KWKRQTAVGLELLAE
8. LBX1    ESPPVRAKKPRKARTAFSDHQLNQLERSFERQKYLSVQDRMDLAAALNLTDTQVK TWYQNRRA KWKRQTAVGLELLAE
9. EMX1    ESPPVRAKKPRKARTAFSDHQLNQLERSFERQKYLSVQDRMDLAAALNLTDTQVK VWYQNRRT KWKRQTAVGLELLAE
10.CDX1    ESPPVRAKKPRKARTAFSDHQLNQLERSFERQKYLSVQDRMDLAAALNLTDTQVK IWYQNRRA KWKRQTAVGLELLAE
11.NANOG   ESPPVRAKKPRKARTAFSDHQLNQLERSFERQKYLSVQDRMDLAAALNLTDTQVK TWYQNRRM KWKRQTAVGLELLAE
12.PKNOX1  ESPPVRAKKPRKARTAFSDHQLNQLERSFERQKYLSVQDRMDLAAALNLTDTQVK NWYQNRRR KWKRQTAVGLELLAE
13.SIX6    ESPPVRAKKPRKARTAFSDHQLNQLERSFERQKYLSVQDRMDLAAALNLTDTQVK NWYQNRRQ KWKRQTAVGLELLAE
14.PBX1    ESPPVRAKKPRKARTAFSDHQLNQLERSFERQKYLSVQDRMDLAAALNLTDTQVK NWYQNRRI KWKRQTAVGLELLAE
```

**Extended Data Fig. 4 | Comparison of BARHL2-DNA^AC structure with the representatives of different homeobox TF families.** Structural alignment of BARHL2-DNA^AC complex (green) with the complexes of anterior HOXB1-DNA^TGAT (blue) and posterior HOXA9-DNA^TTAC (violet) **a**, Structural alignment of BARHL2-DNA^AC complex (green) with UBX-DNA^AT (pink), EXD-DNA^AT (dark green), Clawless-DNA^TAAT (light green) and Aristaless-DNA^TAAT (brown) **b**, Structural alignment of BARHL2-DNA^AC complex (green) with LHX4-DNA^AT (yellow), MEIS1-DNA^TGAC (dark blue) and HOXB13-DNA^TTATT (salmon) **c**, (All structures are solved in Taipale lab). The key residues ASN and ARG as well as the key bases T and A involved in binding are labeled. **d**, Sequence alignment of the structures presented in (**a-c**). The recognition helix is highlighted in yellow, the recognizing residues colored red. The numbering on the top of the sequence corresponds to BARHL2 numbering. **e**, The DNA-binding domain sequences of fourteen mutants designed to test the role of the threonine (Thr278 and Thr285) in the DNA sequence recognition. In the BARHL2 DNA-binding domain sequence we mutated the Thr278 and Thr285 to the corresponding residues of 14 different homeobox proteins. Thr278 and Thr285 are colored green in the original BARHL2 sequence (on the top) and in the mutated sequences if they belong to the original sequence of other homeobox protein. The mutations are colored red.

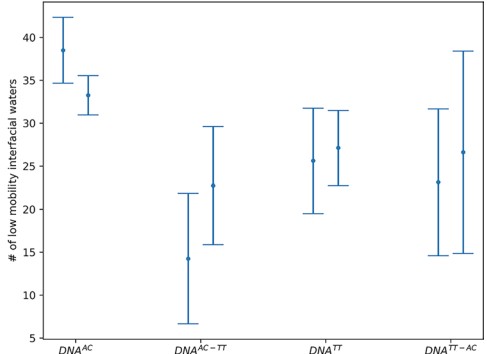

**Extended Data Fig. 5 | Assessment of convergence for the molecular dynamics analysis of water mobility in the protein-DNA interfacial region.** (see main paper Fig. 3j). Data generated by applying the same methodology independently to the first and second halves of each simulation. Absolute numbers of predicted low-mobility waters are higher, due to the shorter sample window, however there is no sign of systematic change in values between the first and second subsamples, and all trends between different molecular systems are maintained. Data are presented as mean values +/- SEM. For DNA$^{AC}$ and DNA$^{AC\text{-}TT}$ the sample size is 4 (dynamically independent copies of the protein-DNA complex within the crystallographic unit cell), while similarly, for DNA$^{TT}$ and DNA$^{TT\text{-}AC}$, the sample size is 8.

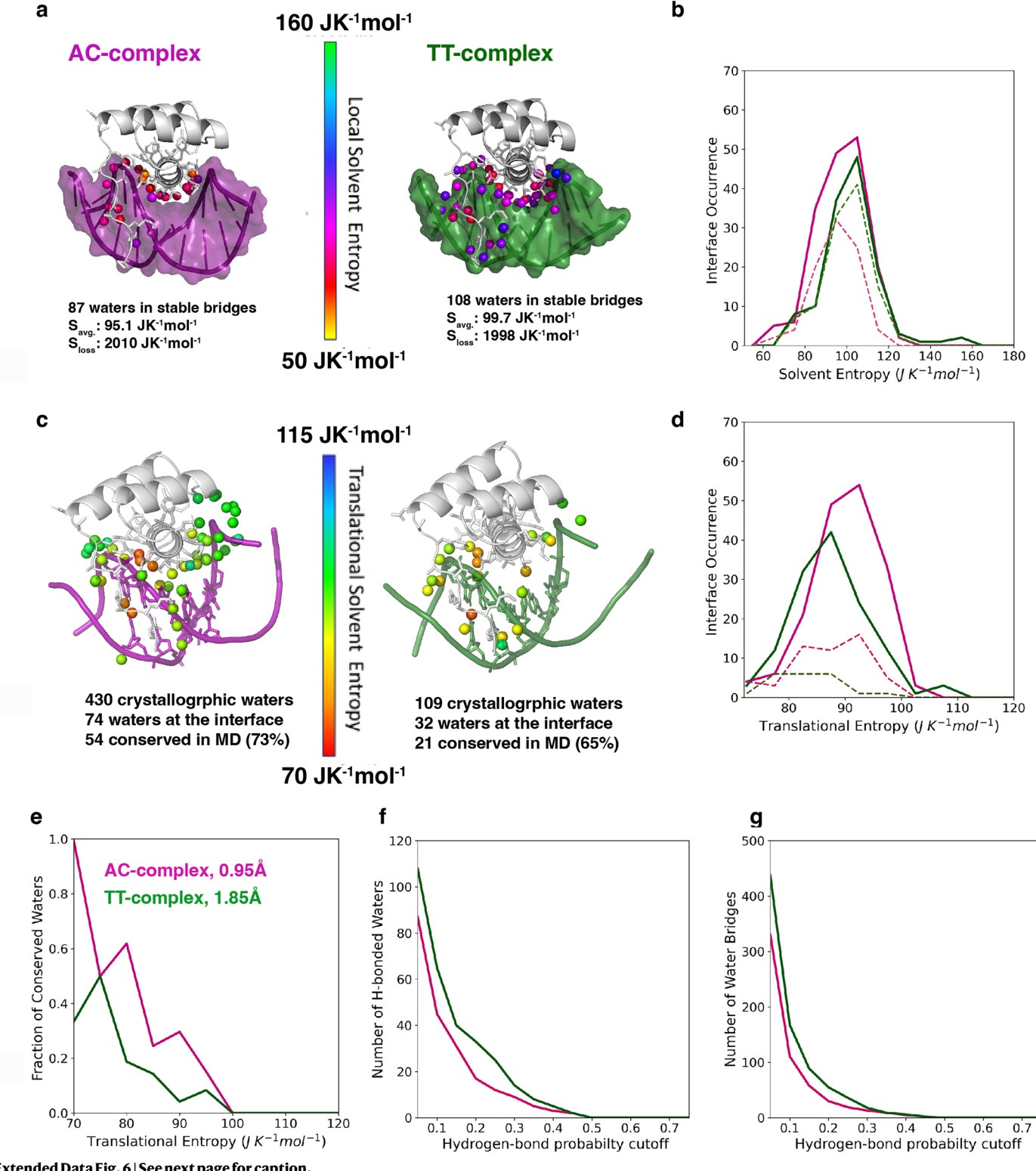

**Extended Data Fig. 6 | See next page for caption.**

**Extended Data Fig. 6 | Further analysis of interfacial water. a**, Water molecules in BARHL–DNA$^{AC}$ (purple) and BARHL2-DNA$^{TT}$ (green) complex interfaces involved in stable, water-mediated hydrogen bond networks connecting protein (gray) amino acids to DNA nucleotides. The waters (spheres) are colored according to their solvent entropy. The number of water molecules in water-bridge networks, as well as the average solvent entropy ($S_{avg}$) of these waters, and their estimated total solvent entropy loss compared to bulk ($S_{loss}$) are shown below. **b**, Solvent entropy distribution of all interfacial waters (solid lines) and the solvent entropy distribution of waters involved in water mediated hydrogen bonding (dashed lines). **c**, Conserved water positions (spheres) representing low mobility water positions at the BARHL2-DNA interface that were also observed in the crystallographic structures. The color of the waters indicates their first order translational entropy in MD simulations. The total number of observed crystallographic waters within 1 nm of the complex, the number crystallographic waters at BARHL2-DNA interface, and the number of conserved water positions in MD simulations are shown below. **d**, The distribution of the translational solvent entropy of water molecules (solid lines) in the interface of the BARHL2:DNA$^{AC}$ and BARHL–DNA$^{TT}$ complexes. Occurrence of all (solid lines) and conserved (dashed lines) interfacial waters with a given translational entropy in MD simulations are shown. **e**, The fraction of conserved interfacial waters in the crystal structures of the BARHL2–DNA$^{AC}$ and BARHL2–DNA$^{TT}$ divided by all interfacial waters in MD simulations as the function of translational entropy of water (from values in **d**). Translational entropy of crystallographic interfacial water molecules were estimated from MD water with the nearest mean position within 2 Å. Top right: best achieved resolution of the respective crystal structures. Note that low-mobility water positions can be more readily identified from the electron density, but the identification of low mobility waters may be limited by the crystallographic resolution. **f, g**, Changes in the number of stable hydrogen bonds (**f**) and the number of waters participating in hydrogen bond networks (**g**) as a function of a cutoff value (minimum fraction of time, hydrogen bond networks exist between protein amino acids and DNA bases during MD simulations).

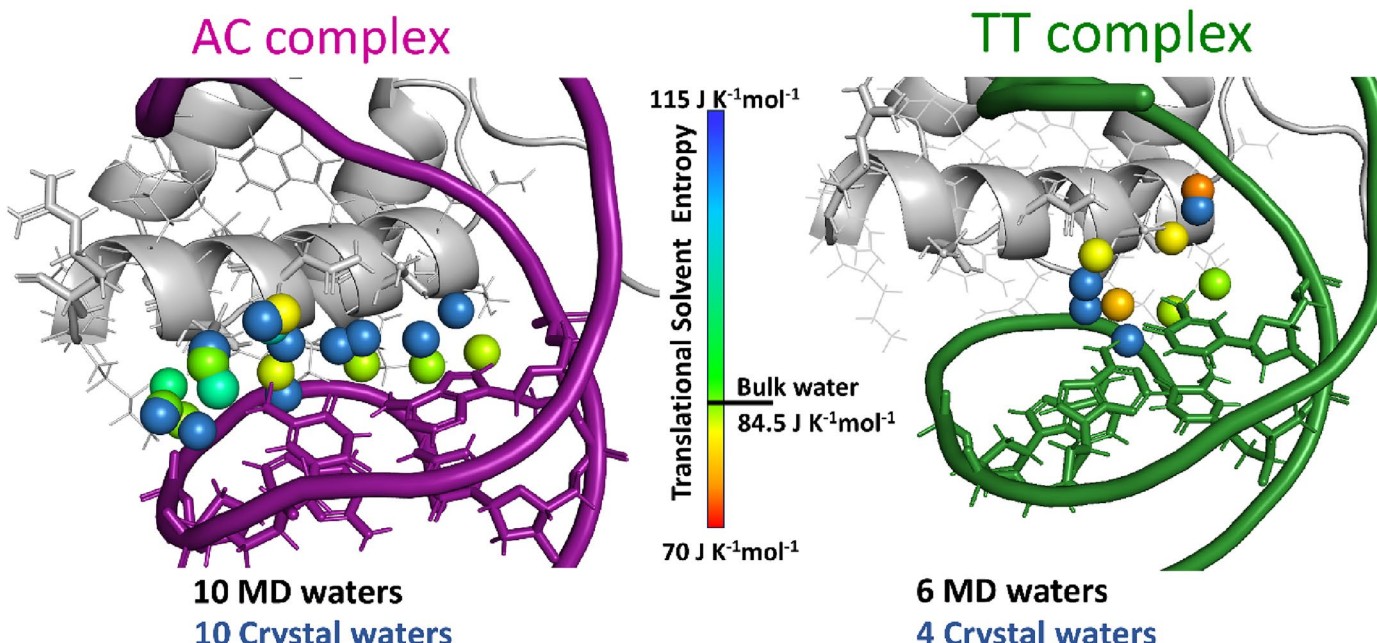

**Extended Data Fig. 7 | Water molecules at BARHL2-variable dinucleotide interface.** DNA for the **BARHL2–DNA[AC]** (purple) and **BARHL2–DNA[TT]** (dark green) complexes are shown as cartoon representations for the backbone and the variable base pairs as sticks. BARHL2 backbone (cartoon) and sidechains (sticks) are shown in gray. The average position of water molecules at the dinucleotide interface in molecular dynamics (MD) simulations is shown as spheres colored according to their estimated translational entropy. Crystallographic water positions for the two BARHL2–DNA complexes are shown as light blue spheres. The number of water molecules around the variable dinucleotides are shown below the image in black (MD) and blue (crystal structure) respectively.

**Extended Data Table 1 | Hydrogen bond distances found in BARHL2-DNA complexes presented on Fig. 3 as dash-lines**

| | T285 Og1– water mol. (Å) | N282Od1 – water mol. (Å) | N282Nd2 –water mol. (Å) | T278Og1 – water mol. (Å) | Dinucleotides – water mol. (Å) | Water mol. – water mol. (Å) |
|---|---|---|---|---|---|---|
| BARHL2:DNA^AC | 1 2.7<br>2 3.6 | 1 2.7<br>3 3.0 | - | 4 2.8<br>5 2.8 | $G_{17}N7$ - 2 2.8<br>$G_{17}O6$ - 6 2.8<br>$T_{18}O4$ - 1 2.7<br>$T_{18}O2$ - 7 2.9<br>$A_7N6$ - 3 3.0<br>$A_7N7$ - 8 2.7 | 2 - 6 2.7<br>6 - 9 2.8<br>9 - 10 2.7<br>10 - 5 2.7<br>3 - 8 2.9<br>8 - 4 2.8<br>3 - 6 2.8 |
| BARHL2:DNA^TT | 1 2.7 | 1 2.8<br>2 3.4 | - | 1 3.5 | $T_8O4$ - 3 2.6<br>$T_8O2$ - 4 3.6<br>$A_{18}N3$ - 4 2.5 | 2 − 5 2.7<br>5 − 3 2.9 |
| BARHL2:DNA^AT | 1 2.5 | 2 2.7<br>1 3.0 | 2 3.6 | | $A_7N6$ - 2 3.0<br>$T_8O2$ - 3 2.8<br>$T_{18}O2$ - 3 2.8<br>$T_{18}O4$ - 1 2.9 | - |
| BARHL2:DNA^TC | 1 2.7 | 1 2.8<br>2 3.0 | - | - | $T_7O4$ - 2 2.8<br>$T_7O4$ - 3 3.3<br>$C_8O2$ - 4 2.9<br>$G_{17}N7$ - 5 2.8<br>$G_{17}N7$ - 3 2.7<br>$G_{17}N2$ - 4 3.6<br>$A_{18}N3$ - 4 2.7 | 1 - 5 3.4<br>5 - 3 2.9<br>3 - 2 3.0 |
| BARHL2:DNA^TG | 1 2.7 | 1 2.8<br>2 2.9 | - | 3 3.4 | $C_5N4$ - 2 3.1<br>$A_6N7$ - 4 2.9<br>$A_6N6$ - 1 3.3<br>$A_6N3$ - 5 2.7<br>$T_7O2$ - 6 2.6<br>$T_7O4$ - 2 2.7<br>$G_8O6$ - 7 2.7<br>$G_8N7$ - 7 2.7 | 1 − 4 2.7<br>2 - 3 2.7<br>3 - 8 2.7<br>5 − 9 2.7<br>9 − 6 3.0<br>7 - 8 2.7 |
| BARHL2:DNA^GT | 1 2.8 | 1 2.7<br>2 2.8 | - | 3 2.8 | $A_5N7$ - 8 2.6<br>$C_6N4$ - 1 2.9<br>$C_6O2$ - 4 2.7<br>$G_7O6$ - 2 2.7<br>$G_7N7$ - 5 2.7<br>$T_8O2$ - 6 2.4 | 6 - 4 2.6<br>2 - 7 3.4<br>2 - 5 2.9<br>3 - 5 2.1<br>8 − 7 2.5<br>8 − 1 3.0 |
| BARHL2:DNA^GC | 1 2.8 | 1 3.3 | - | - | $C_6N4$ 1 3.0 | - |
| BARHL2:DNA^CC | 1 3.4 | 1 3.8 | - | | $G_{18}O6$ 1 3.1 | - |

The water molecules involved in the interactions are numbered in respect of Fig. 3. All distances are presented in Angstrom (Å). The first colomn represents the H-bonds formed between the oxygen atom Og1 of the side chain of Thr285 and the water molecules, the second column represents the H-bond formed between the oxygen atom Od1 of Asn282 and the water molecules, the third colomn shows the H-bonds formed between the nitrogen atom Nd2 and the water molecules; the forth colomn represents the H-bonds formed between the oxygen atom Og1 of Thr278 and the water molecules; the fifth colomn presents the H-bonds formed between varied dinucleotide and the water molecules; the last colomn representes the bridges formed between the water molecules.

# Reporting Summary

## Statistics

For all statistical analyses, confirm that the following items are present in the figure legend, table legend, main text, or Methods section.

| n/a | Confirmed | |
|---|---|---|
| ☐ | ☒ | The exact sample size (*n*) for each experimental group/condition, given as a discrete number and unit of measurement |
| ☐ | ☒ | A statement on whether measurements were taken from distinct samples or whether the same sample was measured repeatedly |
| ☒ | ☐ | The statistical test(s) used AND whether they are one- or two-sided<br>*Only common tests should be described solely by name; describe more complex techniques in the Methods section.* |
| ☒ | ☐ | A description of all covariates tested |
| ☒ | ☐ | A description of any assumptions or corrections, such as tests of normality and adjustment for multiple comparisons |
| ☐ | ☒ | A full description of the statistical parameters including central tendency (e.g. means) or other basic estimates (e.g. regression coefficient) AND variation (e.g. standard deviation) or associated estimates of uncertainty (e.g. confidence intervals) |
| ☒ | ☐ | For null hypothesis testing, the test statistic (e.g. *F*, *t*, *r*) with confidence intervals, effect sizes, degrees of freedom and *P* value noted<br>*Give P values as exact values whenever suitable.* |
| ☒ | ☐ | For Bayesian analysis, information on the choice of priors and Markov chain Monte Carlo settings |
| ☒ | ☐ | For hierarchical and complex designs, identification of the appropriate level for tests and full reporting of outcomes |
| ☒ | ☐ | Estimates of effect sizes (e.g. Cohen's *d*, Pearson's *r*), indicating how they were calculated |

*Our web collection on statistics for biologists contains articles on many of the points above.*

## Software and code

Policy information about availability of computer code

| | |
|---|---|
| Data collection | Crystallographic data were collected using the software developed in synchrotron beam-line ID23-1 in ESRF and listed in Material and Method section |
| Data analysis | Crystallographic data analysis: XDS and CCP4 suits 7.1 and 8.0; MR and refinement: Phaser and Refmac5 as implemented in CCP4 and Phenix.refine; Model building: Coot (versions 0.9.6 and o.9.8.92 (EL)) as implemented in CCP4 and Phenix; Structural visualisation: PyMol 2.5.4; HT-SELEX data analysis: spacek40 (http://github.com/jttoivon/moder2/blob/master/myspacek40.c); Molecular Dynamic simulations in crystal lattice and analysis: AMBER21, UCSF Chimera, Amber 14SB force field and BSC1 parameters for DNA, TIP3P and Joung-Cheatham for water and ions parameters, pmemd.cuda, cpptraj tool, MDTraj, MAtplotlib; Multiple Sequence alignment: Clustal Omega (https://www.ebi.ac.uk/jdispatcher/); Molecular Dynamic simulations for analysis of entropy: GROMACS 2019 simulation package with LINCS algorithm implemented into GROMACS 2019, AMBER-19SB forcefield with OL15 modified parameters for DNA, Per\|Mut for calculations of spatially resolved solvent entropies (https://gitlab.gwdg.de/lheinz/hydration_entropy), Python 3.11.8 and modules: NumPy 1.26.4, MDAnalysis  2.7.0, Matplotlib 3.8.4, and Pandas 2.2.2 |

For manuscripts utilizing custom algorithms or software that are central to the research but not yet described in published literature, software must be made available to editors and reviewers. We strongly encourage code deposition in a community repository (e.g. GitHub). See the Nature Portfolio guidelines for submitting code & software for further information.

## Data

Policy information about availability of data

All manuscripts must include a data availability statement. This statement should provide the following information, where applicable:

- Accession codes, unique identifiers, or web links for publicly available datasets
- A description of any restrictions on data availability
- For clinical datasets or third party data, please ensure that the statement adheres to our policy

Crystal structures were deposited to the protein data bank (PDB) with accession codes: 7Z5I and 7Z5K for crystal structures of MYF5. The PDB code 1MDY was used for the Molecular replacement. The accession codes 8PMF, 8PMN, 8PMC, 8PM5, 8PM7, 8PMV, 8PN4, 8PNA and 8PNC are for BARHL2 structures. The accession code 3A01 was used for structure determination. The details are presented in Table 1. All sequence reads are deposited to the European Nucleotide Archive under the accession number PRJEB65950. The DNA ligands used in the Temperature HT-SELEX experiments are presented in Extended Data Table 3. The details are described in the Material& Method section.

## Research involving human participants, their data, or biological material

Policy information about studies with human participants or human data. See also policy information about sex, gender (identity/presentation), and sexual orientation and race, ethnicity and racism.

| | |
|---|---|
| Reporting on sex and gender | This study did not involve human participants |
| Reporting on race, ethnicity, or other socially relevant groupings | This study did not involve human participants |
| Population characteristics | This study did not involve human participants |
| Recruitment | This study did not involve human participants |
| Ethics oversight | Ethics permissions were not applicable to this study |

Note that full information on the approval of the study protocol must also be provided in the manuscript.

# Field-specific reporting

Please select the one below that is the best fit for your research. If you are not sure, read the appropriate sections before making your selection.

☒ Life sciences          ☐ Behavioural & social sciences          ☐ Ecological, evolutionary & environmental sciences

For a reference copy of the document with all sections, see nature.com/documents/nr-reporting-summary-flat.pdf

# Life sciences study design

All studies must disclose on these points even when the disclosure is negative.

| | |
|---|---|
| Sample size | Our study did not perform a formal sample size calculation. For solving crystal structures: a few dozens crystallization conditions were tested and at least 50 crystals of each complex were tested at the synchrotron beam-line to find the best diffraction. The data sets were collected from one crystal of each complex. For mutational analysis 14 single and double mutants were designed, synthesized and added to the HT-SELEX experiments. |
| Data exclusions | No data were excluded from the analysis |
| Replication | The X-ray data were collected from a single crystals of each complex. The structure of BARHL2/DNAAC was solved at two different resolution to be sure that the 0.95 Å resolution structure is the same as the structure at 1.3 Å resolution. The statistics of data collections and refinements are presented in Table 1. The DNA motif data were obtained from the multiple cycles of SELEX experiments. A total of four SELEX cycles were performed. The DNA ligands from 0, 3rd, and 4th SELEX cycles were sequenced. In the experiments with Temperature SELEX the ligands from each cycle of four cycles and the input were sequenced and analyzed. The conclusions drawn were supported by all SELEX cycles and/or replicates. |
| Randomization | No grouping was involved in the experiments - no randomization was conducted as a result. |
| Blinding | There were no groups or human or animal participants involved in the experiments - no blinding was required |

# Reporting for specific materials, systems and methods

We require information from authors about some types of materials, experimental systems and methods used in many studies. Here, indicate whether each material, system or method listed is relevant to your study. If you are not sure if a list item applies to your research, read the appropriate section before selecting a response.

## Materials & experimental systems

| n/a | Involved in the study |
|-----|----------------------|
| ☒ | Antibodies |
| ☒ | Eukaryotic cell lines |
| ☒ | Palaeontology and archaeology |
| ☒ | Animals and other organisms |
| ☒ | Clinical data |
| ☒ | Dual use research of concern |
| ☒ | Plants |

## Methods

| n/a | Involved in the study |
|-----|----------------------|
| ☒ | ChIP-seq |
| ☒ | Flow cytometry |
| ☒ | MRI-based neuroimaging |

## Plants

| | |
|---|---|
| Seed stocks | no |
| Novel plant genotypes | no |
| Authentication | no |

