## [Peer Review File · Nature Structural & Molecular Biology]

Interfacial water confers transcription factors with dinucleotide specificity

Corresponding Author: Professor Jussi Taipale

Version 0:

Decision Letter:

5th Jan 2024

Dear Professor Taipale,

I am writing you on behalf of my colleague Dr Sara Osman, that is currently out of the office.

Thank you again for submitting your manuscript "Interfacial water confers transcription factors with dinucleotide specificity". I apologize for the delay in responding, which resulted from the difficulty in obtaining suitable referee reports. Nevertheless, we now have comments (below) from the 2 reviewers who evaluated your paper. In light of those reports, we remain interested in your study and would like to see your response to the comments of the referees, in the form of a revised manuscript.

You will see that while both reviewers appreciate the work, reviewer #1 finds that the study fails to connect adequately to the mass of published literature on the subject, and may not fully consider other contributors to thermodynamic readouts than ordered waters (such as conformational changes in protein elements related to binding). Additionally, reviewer #2 echoes the need to place the findings more appropriately in the context of the literature, and asks to better clarify and quantify the influence of DNA shape, suggests to make the dinucleotide mechanism better substantiated by using a high throughput method such as SELEX-seq to show that a binding affinity change can be predicted based on the proposed mechanism, and to further validate the role of the water molecules by running simulations in the presence and absence of water. Editorially, we agree that strengthening these aspects would be necessary to enhance the impact of the study.

We appreciate the requested revisions are extensive. We thus expect to see your revised manuscript within 6 months. If you cannot send it within this time, please let us know. We will be happy to consider your revision as long as nothing similar has been accepted for publication at NSMB or published elsewhere. Should your manuscript be substantially delayed without notifying us in advance and your article is eventually published, the received date would be that of the revised, not the original, version.

Reporting Summary:

Please note that all key data shown in the main figures as cropped gels or blots should be presented in uncropped form, with molecular weight markers. These data can be aggregated into a single supplementary figure. While these data can be displayed in a relatively informal style, they must refer back to the relevant figures. These data should be submitted with the last revision, prior to acceptance, but you may want to start putting it together at this point.

We require deposition of coordinates (and, in the case of crystal structures, structure factors) into the Protein Data Bank with the designation of immediate release upon publication (HPUB). Electron microscopy-derived density maps and coordinate data must be deposited in EMDB and released upon publication. Deposition and immediate release of NMR chemical shift assignments are highly encouraged. Deposition of deep sequencing and microarray data is mandatory, and the datasets must be released prior to or upon publication. To avoid delays in publication, dataset accession numbers must be supplied with the final accepted manuscript and appropriate release dates must be indicated at the galley proof stage. Please find the complete NRG policies on data availability at <http://www.nature.com/authors/policies/availability.html>.

Link Redacted

Sincerely,

Carolina Perdigoto, PhD
Chief Editor
Nature Structural & Molecular Biology
orcid.org/0000-0002-5783-7106

Referee expertise:

Referee #1: X-ray crystallography, protein-nucleic acid interactions

Referee #2: Molecular biophysics of protein-nucleic acid interactions

Reviewers' Comments:

Reviewer #1:

Remarks to the Author:

The article "Interfacial water confers transcription factors with dinucleotide specificity" by Morgunova et al. describes a structural analysis (relying on a series of high resolution protein-DNA cocrystal structures of two transcription factors (the MYF5 bHLH protein, and the BARHL2 homeodomain protein), each in complex with multiple DNA targets (representing variable affinity and differing thermodynamic signatures of binding) to study the physical basis of dinucleotide recognition specificity. The authors' primary conclusion from this study is offered both in the abstract ("water-mediated recognition is important in predicting affinities of macromolecules from their sequence") and in the discussion "dinucleotide recognition critically depends on water molecules in the protein-DNA interface".

The study consists of:

1. Structures of MYF5 bound to two different DNA targets (differing by a single dinucleotide that flanks one side of the protein's E-box core sequence, that results in similar binding affinities and corresponding free energy changes of binding but differing absolute changes in enthalpy and entropy upon binding that 'counter' one another) at 3.1 and 2.3 Å resolution.

--That analysis results in the observation that the protein binds to both DNA sequences with only small differences in overall DNA conformation and with little or no difference in contacts to bases within the E-box or to the DNA backbone; however the protein interacts with the flanking 'AA' or 'GT' dinucleotides differently, via a water-mediated contact between Arg 91 to an A in the former, versus a direct contact between the same Arg and a G in the latter. The authors conclude (page 4, lines 11 to 19) that the enthalpically optimal binding to the former site is the result of that fixed water in the interface, while the entropically optimal binding to the latter site is the result of that water being released and replaced by a direct protein sidechain to base contact.

-- The authors state that while this result and observation is consistent with the different thermodynamic signatures of binding to the two related DNA sites, that it does not explain the dinucleotide (AA or GT) preference displayed by the protein at the flanking positions in the binding target(s).

2. Expansion and broadening of the analysis described above, by then determining structures of BARHL2 (which represents a transcription factor with a quite different structural topology) in complex with eight different DNA targets that include two that represent enthalpically- and entropically- optimized binding affinities, two that differ from those sequences by one basepair, and four more that are 'suboptimal' sequences.

-- A gratifying feature of this analysis, that in and of itself increases the appeal of this study and manuscript, is that this latter comparative analysis includes structures extended to resolutions as high as 0.95 Å; the subset of transcription factor /DNA cocrystal structures solved to that high of resolution (allowing the clear visualization of alternative conformations for the DNA backbone within the complex) will obviously provide computational structural biologists interested in modeling, predicting and/or designing protein-DNA complexes with some very useful new data.

-- Like the prior analysis, the protein binds to its two optimal DNA sequences with only small difference in overall DNA conformation. Also similar to the prior analysis, the protein interacts with its enthalpically-optimal and entropically-optimal 'optimal' sites via quite different reliance on water-mediated contacts: the protein bound to the enthalpic optimal target (TAAACG) displays on an extensive, well-ordered water network surrounding the variable dinucleotide within that target, whereas the same protein bound to the entropic optimal target (TAATTG) instead displays a much reduced number of ordered interfacial water molecules, which are largely replaced by direct (largely hydrophobic van der Waals) contacts between protein and bases (described in detail on pages 5 to 7).

-- The analysis is then augmented by examination of the contacts formed to two additional targets (TAAATG and TAATCG) that represent two single basepair substitutions relative to the optimal targets, that each 'break' the dinucleotides present in the optimal DNA targets, and argue that the structures provide details on the cooperative recognition of distinct dinucleotide sequence elements within the target, rather than simpler 'additive' recognition of individual basepairs within each dinucleotide.

-- The relative importance of the various water-mediated and (hydrophobic) direct contacts observed in the structures summarized above were then further tested / validated by mutation of the residues (Thr 278 and 285) that serve as the 'core' of each protein-DNA recognition interface described above.

--Finally, the temperature-dependence of binding affinity was tested for the enthalpically- and entropically-optimal complexes, with the expectation that they would display different behaviors reflecting the overall balance of thermodynamic drivers of binding.

Comments:

1. Overall, the experimental work described in this study, in particular the crystallographic analyses, is of high technical quality.

2. I had to retrieve reference #19 to examine the underlying calorimetry studies on MYF5 and BARHL2 against what are described as those factor's 'enthalpically optimal' and 'entropically optimal' DNA targets (Figure 5 and corresponding caption and methods in that paper. The quality of those experiments and corresponding computation of ΔH , ΔS and ΔG of binding is unclear to me (and does not appear to have been critically evaluated by that study's original reviewers).

In particular, the isotherms for MYF5 are highly unusual (indicating a release of heat in early injections, then an absorption of heat in later injections, making it appear to this reviewer that the buffers in the reference and test cells, relative to the injectant buffers, were not properly equilibrated with one another).

3. My primary concern with this study, however, is not technical at all (I always appreciate detailed structural studies of how molecular recognition, and variable fidelity of recognition within given molecular complexes, is driven by differences in contacts, and feel that this study is technically quite solid).

Instead, my problem with the paper is conceptual: I feel as though the topic of this study has been the subject of many structural and computational studies throughout the past 20 years – too many published studies to try to list here, but a couple hours in front of PUBMED and Google Scholar will provide any investigator with a large, expansive set of structural and biophysical studies of protein-DNA recognition that assess the role of interfacial water in sometimes subtly (and sometimes significantly) driving protein-DNA recognition), and that this study, while technically quite interesting, fails to adequately connect to that literature.

In this reviewer's opinion (and this is obviously very subjective), the authors' primary conclusion ("water-mediated recognition is important in predicting affinities of macromolecules from their sequence" and "dinucleotide recognition critically depends on water molecules in the protein-DNA interface") is (on the one hand) somewhat incremental (and perhaps rather obvious) relative to the published literature, and (on the other hand) somewhat reductionistic and over-simplified relative to all the physical factors that contribute to changes in enthalpy and entropy that contribute to overall binding affinity and specificity. Obviously, a major contributor to the magnitude of entropic change that accompanies protein-DNA binding is the release and/or retention and re-organization of ordered waters within the DNA target site. However, other factors (such as the balance between binding-induced organization of flexible protein elements, versus pre-organization of other protein elements in the interface) also contribute to the overall magnitude of ΔS .

4. I am not sure exactly how the authors are defining direct versus indirect recognition. I believe that the former refers to sequence-specific readout of base identity, via both direct contacts between protein groups and DNA bases and also water-mediated contacts between protein and DNA (said contacts resulting in stereochemically complementary contacts that may involve both directional hydrogen bonds and nonpolar van der Waals contacts), whereas the latter refers to structural features and deformation of the bound DNA (either induced by the protein upon binding, or sampled by the unbound DNA and then captured by the protein) that have an underlying DNA sequence propensity. I would recommend that the authors clearly define exactly what they understand these terms to mean, relative to their own study.

5. I feel that the authors are relying upon and citing rather poorly-aged studies in the literature surrounding fundamental aspects of the field's current understanding of protein-DNA binding and recognition. For example, the authors' first sentence of their discussion, where they state what 'the currently established view of the binding specificity between protein molecules and DNA' is driven by, is supported by the citation of a review written 14 years ago (Rohs et al. 2009) of protein-DNA recognition. Similarly, their first paragraph of their introduction, where they set up the 'straw man' view of DNA binding being the result of additive recognition of individual sequential DNA bases, corresponds to six studies published between 1987 and 2011. It seems to this reviewer that this study and the data within it would be better served by placing it within the context of the most current possible published views of protein-DNA recognition, which are surely far more nuanced than the studies being cited in this work.

Reviewer #2:

Remarks to the Author:

The study by Morgunov et al. introduces a novel aspect of transcription factor (TF) binding mechanism, suggesting that dinucleotide specificity can be achieved through an interfacial water network. The study also proposes this additional mechanism to address the longstanding question of how a TF can bind sub optimal binding sites. The authors focused on MYF5 and BARHL2 TFs by solving co-crystal structures of these TFs with DNA, and explored how they can recognize DNA dinucleotides using two methods: one using a water network, the other a hydrophobic patch. The authors solved 10 structures of MYF5 and BARHL2 bound to different DNA sequences through X-ray crystallography. Additionally, they conducted temperature sensitivity experiments to validate the enthalpic effects and cytosine methylation to validate the hydrophobic patch mechanism for TF-DNA binding.

The study notably contributes to the field by broadening the discussion of water effects, potentially steering future research directions. However, the manuscript contains several unsubstantiated statements and errors that currently hinder its acceptance. To support their conclusions, the authors should undertake more comprehensive comparisons and validations, strengthen the logic of the manuscript and employ quantitative methods to reinforce their conclusions and hypotheses. The authors might consider addressing the following comments:

Major comments:

1. Some statements in the manuscript lack sufficient support and citations. For example, the claim that "the structural distortion caused by histone octamer or DNA-bending TFs is expected to result in relatively weak dinucleotide preferences" and "many TFs having preferences for particular dinucleotides can bind to DNA without inducing major changes to its canonical B-form" are presented without citations. It is important to note that periodic patterns in dinucleotides have been observed in nucleosome binding sites, and these patterns are often correlated with specific DNA structure. Therefore, the authors should provide appropriate citations and a clearer explanation for their statements to strengthen their statements.

2. Likewise, the role of water in protein-DNA recognition has been widely studied and discussed, and I am not sure that this is reflected in the current form of the manuscript.
3. The examples provided by the authors do not conclusively eliminate the influence of DNA shape. For example, the manuscript states that the DNA shapes of DNAGT and DNAAA bound to MYF5 are very similar, suggesting that structure is not a significant factor in their high affinity binding. However, differences in DNA conformation related to Arg91, as illustrated in Figure 1f and Figure 1g, could suggest an alternate explanation: the specific shape of DNAGT might facilitate the formation of water-mediated hydrogen bonds due to larger accessibility, while the shape of DNAAA is conducive to non-water-mediated contacts due to the limited space that precludes water molecules from being present, which results in direct interaction with Arg91. The authors should address this aspect and provide a more comprehensive explanation of how DNA shape influences or does not influence binding specificity in these scenarios.
4. How is the electrostatic potential calculated? What is the range of values? What dielectric constants did the authors use, and what was the reason for this choice?
5. In Fig 1d, without the x- and y-axis or other context, it is hard to tell what the colored lines represent. The authors should consider adding legends such as in Figures 1d, 2f, and 2h.
6. Moreover, when comparing DNA structure with 3DNA, the authors, from my understanding, purely relied on their objective opinions. It will be helpful to provide a quantitative comparison of DNA shape rather than using only visualization for Figs. 1 and 2. In addition, it seems to me that the DNA backbone and electrostatics in Figs. 1d and 1e are pretty different. The authors need to use more robust statistical analysis to show the sequences have similar backbone shape/electrostatics. Lastly, when the authors perform such analysis, it would be interesting to consider flanking DNA influences.
7. The sequence in Figure 1d appears to be incorrect. It is currently presented as AACAGTCGAC, but it should likely be AACAGCTGAC, and GTCAGTCGAC should be GTCAGCTGAC. Most importantly, the authors should verify that the structures being compared correspond to the correct DNA sequences.
8. The authors assert a significant difference in electrostatic potentials within the major groove, suggesting that DNA backbone shape and electrostatic potential do not account for MYF5's preference for flanking AA and GT sequences which both demonstrate high binding affinity. However, this claim seems inconsistent with the data presented in Figure 1e, which refers to the minor groove where no contact is observed. This discrepancy indicates that the statement may be incorrect.
9. For the MD simulations, did the authors check equilibration of the simulations? 200ns might be too short for a protein-DNA system. Any analysis of the trajectory, such as that in Fig. 3j, should be based on the equilibrated part of the simulation trajectory. Last, did authors run any replicas to confirm the results?
10. The authors should include how they in silico mutated the DNA for simulations in the Methods section.
11. The authors stated that the dinucleotide is the reason there are differences in recognition between sequences rather than shape or electrostatics but the authors only showed that it is true for a few sequences. To make the dinucleotide mechanism more convincing, the authors can show that it is true for more sequences by using high throughput experiments such as SELEX-seq. It would be convincing to see if the authors can really predict a binding affinity change using their proposed mechanism.
12. For enhanced validation of their findings, the authors should consider additional experiments, for example, using molecular dynamics to simulate the studied system both in the presence and absence of water. This will elucidate the role of water in DNA binding dynamics, revealing any significant changes attributable to the water-mediated interactions, or mutating the residues that are involved in water-mediated contacts with DNA. By analyzing the impact of these mutations on DNA binding affinity and specificity, the authors can directly assess the functional importance of these water-mediated interactions.
13. The authors stated: "If BARHL2 would interact with mononucleotides within a sequence motif independently, the highest affinity sequence would be one of the sequences (either DNAAT or DNATC)". Won't the highest affinity sequence still be AC according to the PWM in Fig. 2A? AT/TC may show higher affinity than TT if BARHL2 interacts with mononucleotide but why would they be the highest affinity?

Minor comments:

1. The distance measurements in Fig. 3 are not very visible.
2. The figure quality needs to be improved, especially Figures 4a, b, e, which has low resolution. Authors can consider removing the underscores in their axis labels.
3. In section "Mechanism of BARHL2 binding to dinucleotides", the authors forgot to highlight the dinucleotide for the first of four suboptimal sequences.
4. In Fig. 2g, the authors claimed that "largest shift from B-shape was observed at the position of the variable dinucleotide". However, none of the lines is the B-DNA.
5. The authors should explain more clearly the enrichment score in Figure 3.

In summary, this is a very interesting study with lots of valuable data. The discussion and proof of points put forward are just not where I think it could eventually be to make this a more impactful contribution. I also note that that it is not trivial to just say that this is recognition of dinucleotides. Dinucleotides are both mononucleotides and stacking interaction between them, so sequence and shape, and not something independent or a new entity and basis for novel recognition mode. Water is used to mediate either base or shape readout and its thermodynamic role in readout has been described by many authors over the years. I am hoping this helps and will lead to a revision and eventually publication of the manuscript.

Version 1:

Decision Letter:

Our ref: NSMB-A48351A

26th Jun 2024

Dear Dr. Taipale,

Thank you for submitting your revised manuscript "Interfacial water confers transcription factors with dinucleotide specificity" (NSMB-A48351A). It has now been seen by the original referees and their comments are below. The reviewers find that the paper has improved in revision, and therefore we'll be happy in principle to publish it in Nature Structural & Molecular Biology, pending minor revisions to satisfy the referees' final requests and to comply with our editorial and formatting guidelines.

We are now performing detailed checks on your paper and will send you a checklist detailing our editorial and formatting requirements in the next few weeks. Please do not upload the final materials and make any revisions until you receive this additional information from us.

To facilitate our work at this stage, it is important that we have a copy of the main text as a word file. If you could please send along a word version of this file as soon as possible, we would greatly appreciate it; please make sure to copy the NSMB account (cc'ed above).

Sincerely,
Sara

Sara Osman, Ph.D.
Senior Editor
Nature Structural & Molecular Biology

Reviewer #1 (Remarks to the Author):

I am satisfied that the authors have addressed my comments (which were largely regarding two points: up-to-date citation and description of the state of the field in protein-DNA recognition, and a clearer depiction of the novelty and contribution of this study relative to the state of the field) in a good-faith manner. As mentioned previously, the technical quality of this work is extremely high, and certainly the issue of the role of non-covalently associated solvent molecules (and counter-ions) in protein-DNA recognition remains of considerable importance and interest. This is particularly true in this new age of computational structure prediction and docking, and the use of machine-learning algorithms to that end that rely on the availability of high quality, empirically determined structures.

Reviewer #2 (Remarks to the Author):

The authors have addressed all my comments and suggestions in the revision of this manuscript. The work is an example where protein-DNA readout can be described using dinucleotides primarily because the underlying mechanism is structure based.

In this context, I disagree with the other reviewer who labeled a paper that successfully established DNA shape readout and is cited more than 1k times as poorly aged. In fact, the new findings in this manuscript further support that dinucleotides form shape that is in turn recognized by transcription factors.

Lastly, I must apologize for my earlier suggestion to use molecular dynamics (MD) to probe the effect of water. What I had in mind, but did not correctly describe, were comparisons with molecular simulations (MD or Monte Carlo) where water or the solvent is described implicitly.

All my comments and the comments of the other reviewer have been adequately addressed, and I recommend publication of the current manuscript.

Version 2:

Decision Letter:

12th Nov 2024

Dear Dr. Taipale,

We are now happy to accept your revised paper "Interfacial water confers transcription factors with dinucleotide specificity" for publication as a Article in Nature Structural & Molecular Biology.

Your paper will be published online soon after we receive proof corrections and will appear in print in the next available issue. You can find out your date of online publication by contacting the production team shortly after sending your proof corrections.

Please note that *Nature Structural & Molecular Biology* is a Transformative Journal (TJ). Authors may publish their research with us through the traditional subscription access route or make their paper immediately open access through payment of an article-processing charge (APC). Authors will not be required to make a final decision about access to their article until it has been accepted. [Find out more about Transformative Journals](https://www.springernature.com/gp/open-research/transformative-journals)

Authors may need to take specific actions to achieve [compliance](https://www.springernature.com/gp/open-research/funding/policy-compliance-faqs) with funder and institutional open access mandates. If your research is supported by a funder that requires immediate open access (e.g. according to [Plan S principles](https://www.springernature.com/gp/open-research/plan-s-compliance)) then you should select the gold OA route, and we will direct you to the compliant route where possible. For authors selecting the subscription publication route, the journal's standard licensing terms will need to be accepted, including a

[self-archiving policies](https://www.springernature.com/gp/open-research/policies/journal-policies). Those licensing terms will supersede any other terms that the author or any third party may assert apply to any version of the manuscript.

Kind regards,
Florian

Dr Florian Ullrich
Senior Editor, Nature
Consulting Editor, Nature Structural & Molecular Biology
ORCID 0000-0002-1153-2040

Reviewer 1.

Remarks to the Author:

The article “Interfacial water confers transcription factors with dinucleotide specificity” by Morgunova et al. describes a structural analysis (relying on a series of high resolution protein-DNA cocrystal structures of two transcription factors (the MYF5 bHLH protein, and the BARHL2 homeodomain protein), each in complex with multiple DNA targets (representing variable affinity and differing thermodynamic signatures of binding) to study the physical basis of dinucleotide recognition specificity. The authors’ primary conclusion from this study is offered both in the abstract (“water-mediated recognition is important in predicting affinities of macromolecules from their sequence”) and in the discussion “dinucleotide recognition critically depends on water molecules in the protein-DNA interface”.

The study consists of:

This section describes our study, the text is omitted for brevity.

Comments:

1. Overall, the experimental work described in this study, in particular the crystallographic analyses, is of high technical quality.

We thank the reviewer for this positive comment.

2. I had to retrieve reference #19 to examine the underlying calorimetry studies on MYF5 and BARHL2 against what are described as those factor’s ‘enthalpically optimal’ and ‘entropically optimal’ DNA targets (Figure 5 and corresponding caption and methods in that paper. The quality of those experiments and corresponding computation of ΔH , ΔS and ΔG of binding is unclear to me (and does not appear to have been critically evaluated by that study’s original reviewers). In particular, the isotherms for MYF5 are highly unusual (indicating a release of heat in early injections, then an absorption of heat in later injections, making it appear to this reviewer that the buffers in the reference and test cells, relative to the injectant buffers, were not properly equilibrated with one another.

This comment refers to an earlier work, motivating the current study. We want to clarify that we are and were at the time very well aware of the pitfalls of using ITC (see Chodera and Mobley, *Ann Rev Biophys* 42:121-142, 2013). Namely, as entropy is not directly measured, and instead derived from binding constant and enthalpy, any errors in measurement will show up as entropy-enthalpy compensation. For this reason, we used identical buffers for protein and DNA, and observed similar isotherms in replicate experiments. We believe that the isotherm is complex because of the complexity of the binding reaction itself, as it involves two protein molecules and double-stranded DNA (protein has to dimerize, and bind to DNA). Regardless of the source of the complexity, the purpose of the *eLife* work was not to analyse the details of the binding reaction *per se*, but to compare the isotherms of two optimal DNA sequences (in the experiments, only the DNA sequence was varied).

Furthermore, in this work, we have used several methods to validate the enthalpic and entropic optima (structural analysis, molecular dynamics, and temperature SELEX, all of which are consistent with the original ITC results). **We have now added references to papers**

discussing entropy of bound water and entropy-enthalpy compensation (Dunitz, 1994; Chodera and Mobley, 2013 and Fox et al., 2018), and clarified the evidence supporting enthalpic and entropic optima of TF-DNA binding in the results and discussion sections (p. 2, lines 32-33 and p. 14, line 26).

3. My primary concern with this study, however, is not technical at all (I always appreciate detailed structural studies of how molecular recognition, and variable fidelity of recognition within given molecular complexes, is driven by differences in contacts, and feel that this study is technically quite solid).

Instead, my problem with the paper is conceptual: I feel as though the topic of this study has been the subject of many structural and computational studies throughout the past 20 years – too many published studies to try to list here, but a couple hours in front of PUBMED and Google Scholar will provide any investigator with a large, expansive set of structural and biophysical studies of protein-DNA recognition that assess the role of interfacial water in sometimes subtly (and sometimes significantly) driving protein-DNA recognition), and that this study, while technically quite interesting, fails to adequately connect to that literature.

In this reviewer's opinion (and this is obviously very subjective), the authors' primary conclusion ("water-mediated recognition is important in predicting affinities of macromolecules from their sequence" and "dinucleotide recognition critically depends on water molecules in the protein-DNA interface") is (on the one hand) somewhat incremental (and perhaps rather obvious) relative to the published literature, and (on the other hand) somewhat reductionistic and over-simplified relative to all the physical factors that contribute to changes in enthalpy and entropy that contribute to overall binding affinity and specificity. Obviously, a major contributor to the magnitude of entropic change that accompanies protein-DNA binding is the release and/or retention and re-organization of ordered waters within the DNA target site. However, other factors (such as the balance between binding-induced organization of flexible protein elements, versus pre-organization of other protein elements in the interface) also contribute to the overall magnitude of ΔS .

We agree that the importance of water in protein-DNA recognition has previously been reported. However, the importance of water to the recognition of dinucleotides has not, to our knowledge, been shown before. **We have now made this distinction clearer by citing several seminal papers showing water-mediated DNA recognition (Otwinowski et al., 1988; Rodgers and Harrison, 1993; and a review by Schwabe, 1997), and by rewriting the relevant parts of the Introduction (p. 2, lines 6-8) and Discussion (p. 14, lines 3-5 and 26-28) sections of the manuscript. We have also added a note to the Acknowledgements (page 17) to apologise to the authors of other structural studies of protein-DNA complexes that we cannot unfortunately cite due to space limitations of the journal.**

Although many have structurally studied DNA-binding of mutant proteins and closely related proteins to DNA, we are not aware of studies that have as extensively analysed binding of TFs to different DNA sequences at high resolution. **We have also now clarified this aspect in the discussion (p. 15, line 31).**

We also of course agree that protein contributes to entropy. To address this point, **we have now performed extensive molecular dynamics simulations to evaluate the contributions of protein, DNA and solvent to binding entropy of the enthalpic (AC) and entropic (TT) optima of BARHL2 (new Figure 5 panels a-e).** These results show that contribution of the protein and DNA to entropy is lower than that of the solvent waters, supporting our model of solvent contribution to dinucleotide recognition.

4. *I am not sure exactly how the authors are defining direct versus indirect recognition. I believe that the former refers to sequence-specific readout of base identity, via both direct contacts between protein groups and DNA bases and also water-mediated contacts between protein and DNA (said contacts resulting in stereochemically complementary contacts that may involve both directional hydrogen bonds and nonpolar van der Waals contacts), whereas the latter refers to structural features and deformation of the bound DNA (either induced by the protein upon binding, or sampled by the unbound DNA and then captured by the protein) that have an underlying DNA sequence propensity. I would recommend that the authors clearly define exactly what they understand these terms to mean, relative to their own study.*

We apologise for the unclear definition of the direct and indirect interactions. We are aware that usage of the terms varies between publications, so we have now stated (p. 2, line 13, p. 15. line 1) that for clarity, in our work we use the word "direct" in only one capacity, to mean molecular interactions that involve direct contact between amino-acid and DNA base), and use the word "indirect" for any other means of recognition that is not direct, for example interactions between via water-mediated hydrogen bonds, or by bending of DNA etc.).

5. *I feel that the authors are relying upon and citing rather poorly-aged studies in the literature surrounding fundamental aspects of the field's current understanding of protein-DNA binding and recognition. For example, the authors' first sentence of their discussion, where they state what 'the currently established view of the binding specificity between protein molecules and DNA' is driven by, is supported by the citation of a review written 14 years ago (Rohs et al. 2009) of protein-DNA recognition. Similarly, their first paragraph of their introduction, where they set up the 'straw man' view of DNA binding being the result of additive recognition of individual sequential DNA bases, corresponds to six studies published between 1987 and 2011. It seems to this reviewer that this study and the data within it would be better served by placing it within the context of the most current possible published views of protein-DNA recognition, which are surely far more nuanced than the studies being cited in this work.*

We thank the reviewer for pointing this out. We agree of course that it is well known that the mononucleotide models are incorrect and only approximate. Our intent was not to set up a straw man, but to highlight the fact that even if on average PWMs are often useful approximations and still in common use, some TFs will very strongly deviate from a PWM model. **We have now rewritten the first paragraphs to clarify this point (p. 2, lines 14-16 and p.14, lines 8-10). In addition, we have added several references to earlier seminal papers about the role of water in TF binding specificity, and also now cite more recent experimental and computational papers (Stormo, 2013; Rastogi et al., 2018; Ge et al. 2021 and Chiu et al., 2023; p. 2, line 15) that clearly show that additive models are in many cases good approximations, but fail to quantitatively predict binding.**

Reviewer #2.

Remarks to the Author:

The study by Morgunova et al. introduces a novel aspect of transcription factor (TF) binding mechanism, suggesting that dinucleotide specificity can be achieved through an interfacial water network. The study also proposes this additional mechanism to address the longstanding question of how a TF can bind sub optimal binding sites. The authors focused on MYF5 and BARHL2 TFs by solving co-crystal structures of these TFs with DNA, and explored how they can recognize DNA dinucleotides using two methods: one using a water network, the other a hydrophobic patch. The authors solved 10 structures of MYF5 and BARHL2 bound to different DNA sequences through X-ray crystallography. Additionally, they conducted temperature sensitivity experiments to validate the enthalpic effects and cytosine methylation to validate the hydrophobic patch mechanism for TF-DNA binding.

The study notably contributes to the field by broadening the discussion of water effects, potentially steering future research directions. However, the manuscript contains several unsubstantiated statements and errors that currently hinder its acceptance. To support their conclusions, the authors should undertake more comprehensive comparisons and validations, strengthen the logic of the manuscript and employ quantitative methods to reinforce their conclusions and hypotheses. The authors might consider addressing the following comments:

Major comments:

1. Some statements in the manuscript lack sufficient support and citations. For example, the claim that “the structural distortion caused by histone octamer or DNA-bending TFs is expected to result in relatively weak dinucleotide preferences” and “many TFs having preferences for particular dinucleotides can bind to DNA without inducing major changes to its canonical B-form” are presented without citations. It is important to note that periodic patterns in dinucleotides have been observed in nucleosome binding sites, and these patterns are often correlated with specific DNA structure. Therefore, the authors should provide appropriate citations and a clearer explanation for their statements to strengthen their statements.

We thank the reviewer for pointing this issue out. We have now clarified that the local effect of DNA bending on dinucleotide preference is small within a short segment of DNA, such as that bound by TFs. In a nucleosome, which binds much longer DNA, the small effects add up to allow sequence to contribute to nucleosome positioning. But a local bend, even as strong as what is seen in a nucleosome, would not cause as strong dinucleotide preference than what is seen in BARHL2 and MYF5, for example. Furthermore, the effect of protein alpha-helix inserting into the major groove of DNA has a much smaller effect on DNA backbone shape than formation of a nucleosome. Therefore, a mechanism other than local DNA bending must account for the dinucleotide specificity of both MYF5 and BARHL2. **We have now clarified this and now cite a review that covers energetic contribution to DNA bending (Biswas and Basu, 2023), and also cite a structural paper (Pavletich and Pabo, 1991) (p. 14, line 20) that shows that a common mode of DNA sequence recognition used by both MYF5 and BARHL2, insertion of a protein alpha-helix to the major groove of DNA, does not result in major distortion of the B-shape of DNA (p. 6, line 26, and Extended Data Fig.2). In addition, we have carefully reviewed the text and added citations to all similar statements.**

2. Likewise, the role of water in protein-DNA recognition has been widely studied and discussed, and I am not sure that this is reflected in the current form of the manuscript.

We agree of course that the role of water in protein-DNA recognition itself is well established, and as described above in response to Reviewer #1, **we have now added references to several seminal papers showing water-mediated DNA recognition (Otwinowski et al., 1988; Rodgers and Harrison, 1993; and a review by Schwabe, 1997). We have also clarified that what is new in our work is the role of water in recognition of dinucleotides (p. 15, lines 6-7).**

3. The examples provided by the authors do not conclusively eliminate the influence of DNA shape. For example, the manuscript states that the DNA shapes of DNAGT and DNAAA bound to MYF5 are very similar, suggesting that structure is not a significant factor in their high affinity binding. However, differences in DNA conformation related to Arg91, as illustrated in Figure 1f and Figure 1g, could suggest an alternate explanation: the specific shape of DNAGT might facilitate the formation of water-mediated hydrogen bonds due to larger accessibility, while the shape of DNAAA is conducive to non-water-mediated contacts due to the limited space that precludes water molecules from being present, which results in direct interaction with Arg91. The authors should address this aspect and provide a more comprehensive explanation of how DNA shape influences or does not influence binding specificity in these scenarios.

We apologise for the confusion. We did not intend to claim that DNA shape or electrostatic effects are not important for protein-DNA recognition. We have now clarified that a model, where the two optimal sequences are bound equally well because they would have similar DNA shape or charge distribution despite their different DNA sequences, is not supported by our findings. **On the contrary, in both the cases of MYF and BARHL2, the shapes and charge distributions of the optimal sequences differ more from each other than the optimal sequences and more weakly bound sites. This is now clarified on p. 6, lines 20-23.**

4. How is the electrostatic potential calculated? What is the range of values? What dielectric constants did the authors use, and what was the reason for this choice?

We apologise for the error, we did not originally calculate electrostatic potential, what was shown was simply the coloring of the charged atoms. **We have now clarified this and refer to positions of charged atoms throughout (p. 4, lines 19 and 25). Furthermore, we also now include prediction of electrostatic potentials (Extended Data Figure 2), and briefly discuss them on p. 4, lines 21-28.**

5. In Fig 1d, without the x- and y-axis or other context, it is hard to tell what the colored lines represent. The authors should consider adding legends such as in Figures 1d, 2f, and 2h.

We apologise for the unclear legend, and **have now added the information requested to legends to Figures 1d, 2f and 2h. In brief, Figure 1d is a magnified image representing the parts of DNAs showing the largest deviation of the experimental DNA from the B-shape DNA (shown in yellow). We have added a key to the figures to indicate the DNA sequences, and added descriptions to the legends to clarify that the colored lines represent DNA backbone shapes of the different DNAs.**

6. Moreover, when comparing DNA structure with 3DNA, the authors, from my understanding, purely relied on their objective opinions. It will be helpful to provide a quantitative comparison of DNA shape rather than using only visualization for Figs. 1 and 2. In addition, it seems to me that the DNA backbone and electrostatics in Figs. 1d and 1e are pretty different. The authors need to use more robust statistical analysis to show the sequences have similar backbone shape/electrostatics. Lastly, when the authors perform such analysis, it would be interesting to consider flanking DNA influences.

We agree, and have now added the calculated shape parameter values for the free (from prediction) and bound DNAs (from structure) as Extended Data Table 1. We also have clarified that the charge distributions in Figure 1e are indeed different, and inconsistent with a model where the optimal sequences are bound equally well due to similar charge distribution. To address the flanking sequence contribution, and the effect of the divergent dinucleotide on the flanking sequence shape parameters, we include shape parameters for the full DNA sequences used in the crystallization.

7. The sequence in Figure 1d appears to be incorrect. It is currently presented as AACAGTCGAC, but it should likely be AACAGCTGAC, and GTCAGTCGAC should be GTCAGCTGAC. Most importantly, the authors should verify that the structures being compared correspond to the correct DNA sequences.

We thank the reviewer for spotting this error. We have now corrected the sequences in Figure 1d.

8. The authors assert a significant difference in electrostatic potentials within the major groove, suggesting that DNA backbone shape and electrostatic potential do not account for MYF5's preference for flanking AA and GT sequences which both demonstrate high binding affinity. However, this claim seems inconsistent with the data presented in Figure 1e, which refers to the minor groove where no contact is observed. This discrepancy indicates that the statement may be incorrect.

We apologise for the error; we now show the major groove side. We also clarify that what is shown is positions of charged atoms (not electrostatic potential), and that their distribution definitely contributes to DNA recognition. However, the positions of charged atoms between the two optimal sequences are different, indicating that a similarity in charge distribution is not a likely explanation for the high affinity of the AA and GT flanking dinucleotides. Instead, it is explained (in part) by direct binding of G in GT, and water-mediated binding of one A in AA. Why T or alternatively the other A is preferred cannot be easily explained based on these two structures, and that is why we also studied BARHL2. **This is now stated on p. 5, lines 13-15.**

9. For the MD simulations, did the authors check equilibration of the simulations? 200ns might be too short for a protein-DNA system. Any analysis of the trajectory, such as that in Fig. 3j, should be based on the equilibrated part of the simulation trajectory. Last, did authors run any replicas to confirm the results?

We agree and checked the equilibration of the simulations. The numbers of low-mobility waters at the protein-DNA interface were also calculated independently for the first and second 100 ns portions of the “production” phase of the simulations and compared. Because the sampling window is shorter, the absolute number of waters whose root-mean-square

fluctuations classify them as ‘low mobility’ is increased, but there is no sign of systematic change in values between the first and second 100 ns subsamples, and all trends between different molecular systems are maintained. We agree with the reviewer that in normal circumstances 100-200 ns of molecular dynamics would not be expected to be long enough for convergence and equilibration, however these simulations take place within the restraints of the crystal lattice which greatly limits mobility. Furthermore, because our simulations are of a complete unit cell, each contains between four and eight independent copies of the protein-DNA complex so we have ‘internal’ replicates from which the significance of observations can be assessed. **We have added text to the manuscript to clarify this point, and included a figure (Extended Data Figure 5) to the the Extended Data to show the results (pages 20-21).**

In addition, we have performed completely independent MD simulations for 20 times longer time (5 μ s) to analyze water entropy. The difference in the number of low mobility interfacial water molecules between AC and TT still persists in these simulations. **These data are now shown in Fig. 5a-e.**

10. The authors should include how they in silico mutated the DNA for simulations in the Methods section.

We have added the explanation of in silico mutations to Method section (p. 19, line 23 and p. 25, line 12).

11. The authors stated that the dinucleotide is the reason there are differences in recognition between sequences rather than shape or electrostatics but the authors only showed that it is true for a few sequences. To make the dinucleotide mechanism more convincing, the authors can show that it is true for more sequences by using high throughput experiments such as SELEX-seq. It would be convincing to see if the authors can really predict a binding affinity change using their proposed mechanism.

We agree that this would be a great achievement. Such an analysis would require building an entirely new AI framework similar to RoseTTAFold2NA. We feel that this is clearly out of scope of the present work. **We have, however, included a new segment into the discussion section suggesting that future work should investigate this very exciting direction (p. 16, lines 3-11).**

12. For enhanced validation of their findings, the authors should consider additional experiments, for example, using molecular dynamics to simulate the studied system both in the presence and absence of water. This will elucidate the role of water in DNA binding dynamics, revealing any significant changes attributable to the water-mediated interactions, or mutating the residues that are involved in water-mediated contacts with DNA. By analyzing the impact of these mutations on DNA binding affinity and specificity, the authors can directly assess the functional importance of these water-mediated interactions.

We agree that additional molecular dynamic simulations are helpful. However, removing water from such simulations results in anomalous results that cannot be clearly interpreted. So instead, we contacted Prof. Grubmuller, whose Per|Mut method can track water molecules and analyse entropic and enthalpic contributions of individual waters and water networks. We performed extensive new simulations using BARHL2 bound to either AC or TT site. These simulations show major differences in the water networks between the enthalpic (AC) and

entropic (TT) optima. **These results are now added to Figure 5, panels a-e, and to Extended Data Figures 6 and 7, and discussed in new paragraphs in the Results (p. 11-13, and Discussion sections (p. 15, lines 6-30).**

13. The authors stated: “If BARHL2 would interact with mononucleotides within a sequence motif independently, the highest affinity sequence would be one of the sequences (either DNAAT or DNATC)”. Won’t the highest affinity sequence still be AC according to the PWM in Fig. 2A? AT/TC may show higher affinity than TT if BARHL2 interacts with mononucleotide but why would they be the highest affinity?

We agree that this section was unclear, and have rewritten it according to the reviewers’ suggestion (p. 8, lines 21-25)

Minor comments:

1. *The distance measurements in Fig. 3 are not very visible.*

We agree. **We removed the distance measures from this figure, and added them to a new Extended Data Table 3.**

2. *The figure quality needs to be improved, especially Figures 4a, b, e, which has low resolution. Authors can consider removing the underscores in their axis labels.*

We have increased the resolution of these images, and removed the axis labels.

3. *In section “Mechanism of BARHL2 binding to dinucleotides”, the authors forgot to highlight the dinucleotide for the first of four suboptimal sequences.*

Corrected

4. *In Fig. 2g, the authors claimed that “largest shift from B-shape was observed at the position of the variable dinucleotide”. However, none of the lines is the B-DNA.*

We have now clarified that yellow line represents ideal B-DNA backbone shape

5. *The authors should explain more clearly the enrichment score in Figure 3.*

We have now clarified the derivation of the enrichment score in the legend to Figure 3 and in the Methods section (p. 19, line 17).

In summary, this is a very interesting study with lots of valuable data. The discussion and proof of points put forward are just not where I think it could eventually be to make this a more impactful contribution. I also note that that it is not trivial to just say that this is recognition of dinucleotides. Dinucleotides are both mononucleotides and stacking interaction between them, so sequence and shape, and not something independent or a new entity and basis for novel recognition mode. Water is used to mediate either base or shape readout and its thermodynamic role in readout has been described by many authors over the years. I am hoping this helps and will lead to a revision and eventually publication of the manuscript.

We thank the reviewer for these kind comments. **We have also now clarified that the situation where a TF has two optimally bound sequences (epistasis) is only possible if dinucleotides or longer sequences are specifically recognized (p. 15, line 6-13).** What we show here is the mechanism for the recognition.

In summary, we thank the reviewers for their constructive comments, response to which has in our opinion substantially improved the manuscript and clarified its presentation. I hope that after the extensive revisions, corrections and clarifications, our manuscript would now be acceptable for publication.